# RECONSTRUCTION-GUIDED POLICY: ENHANCING DECISION-MAKING THROUGH AGENT-WISE STATE CONSISTENCY

**Qifan Liang**[1], **Yixiang Shan**[1], **Haipeng Liu**[1], **Zhengbang Zhu**[2],
**Ting Long**[1†], **Weinan Zhang**[2], **Yuan Tian**[1]
[1] Jilin University, [2] Shanghai Jiao Tong University
{liangqf23,shanyx22,liuhp24}@mails.jlu.edu.cn
{longting,yuantian}@jlu.edu.cn

## ABSTRACT

An important challenge in multi-agent reinforcement learning is partial observability, where agents cannot access the global state of the environment during execution and can only receive observations within their field of view. To address this issue, previous works typically use the dimension-wise state. This state is obtained by applying MLP or dimension-based attention on the global state for decision-making during training and relying on a reconstructed dimension-wise state during execution. However, dimension-wise states tend to divert agent attention to specific features, neglecting potential dependencies between agents, making optimal decisions more difficult. Moreover, the inconsistency between the states used in training and execution further increases additional errors. To resolve these issues, we propose a method called Reconstruction-Guided Policy (RGP) to reconstruct the agent-wise state, which represents information of inter-agent relationships, as input for decision-making during both training and execution. This not only preserves the potential dependencies between agents but also ensures consistency between the states used in training and execution. We conducted extensive experiments on both discrete and continuous action environments to evaluate RGP, and the results demonstrate its superior effectiveness. Our code is public in `https://github.com/Muise4/RGP4/tree/main`

## 1 INTRODUCTION

Cooperative multi-agent reinforcement learning (Cooperative MARL) (Neto, 2005; Bukharin et al., 2024) refers to the scenario where multiple agents work together within an environment towards a common goal through maximizing a global reward. Due to its practical applications in fields like robot collaboration (Orr & Dutta, 2023), intelligent traffic systems (Mushtaq et al., 2023), distributed energy management (Zhu et al., 2022), network resource allocation (Allahham et al., 2022) and virtual simulations (Liang et al., 2022), this area has gained significant attention in recent years.

A typical challenge in multi-agent reinforcement learning is partial observability (Tuyls & Nowé, 2005; Panait & Luke, 2005), where each agent can only access observations within its field of view instead of the global state, which represents the complete information about all agents and the environment. One direct approach to addressing this issue is to reconstruct the global state (Chen et al., 2022; Xu et al., 2024) within a centralized training and decentralized execution framework to enhance decision-making. As shown in Figure 1 (c), these methods input the **dimension-wise state**, which is obtained by applying an MLP or dimension-wise attention on the global state, into the policy during training to obtain actions for interacting with the environment, and then optimize the policy based on the rewards from the environment. Simultaneously, the global state is used to train a reconstructor, which is designed to reconstruct the global state (or the dimension-wise) when it becomes unavailable. During the execution phase, since the global state is inaccessible, the reconstructed state is used as a substitute input for the policy network to generate actions and interact

---

[†]Corresponding author.

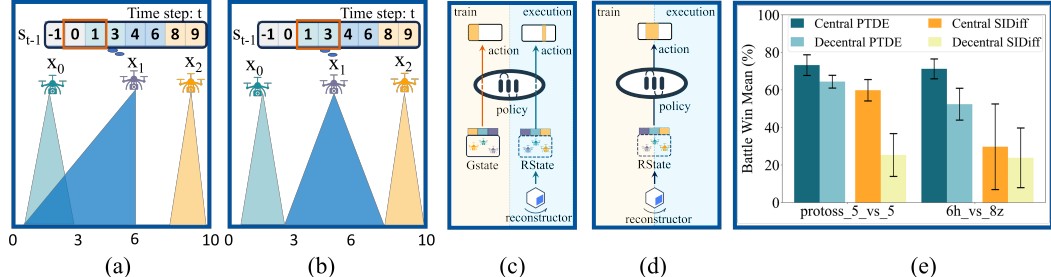

Figure 1: (a) and (b) illustrate a toy example of UAV exploration and rescue. The global state is an 8-dimensional vector: the first two dimensions represent the the state of environment, and the remaining six correspond to the start and end coordinates of the coverage areas for $x_0$, $x_1$, and $x_2$. (c) represents the training and execution process of previous state reconstruction methods, which have inconsistent state input for decision-making. (d) represents the training and execution process of our RGP, which has consistent state input for decision-making. (e) shows the mean win rates of previous state reconstruction methods PTDE (Chen et al., 2022) and SIDiff (Xu et al., 2024) in both centralized and distributed execution.

with the environment. However, these methods have two limitations: (1)State inconsistency. Using a reconstructor to replace the true state with a reconstructed state during execution can introduce additional errors. The policy is trained on the true state, but during execution, it operates on the reconstructed state, which may differ due to reconstruction inaccuracies. This discrepancy reduces the policy's robustness, as it was never trained to handle imperfect states. Therefore, to minimize this error, it is crucial to ensure consistency between the states used during training and execution. (2) Inadequacy of dimension-wise state. The dimension-wise state is a simplified low-dimensional representation of the global state, which may cause the agents to overly focus on specific dimensions rather than on concrete interactive objects. In a toy scenario of UAV search and rescue in Figure 1(a), $x_1$ may overly focus on (0, 1) in the global information instead of the information of $x_0$ (1, 3), leading to overlap with $x_0$ during exploration, as illustrated in the figure. To enable agents to focus on interaction targets rather than specific dimensions, a better approach is to split the global state by agent (as shown in Figure 1 (b)), creating an **agent-wise** state that accounts for inter-agent relationships to improve decision-making.

With that consideration, we propose a method called Reconstruction-Guided Policy (RGP), which consists of two modules: the decision module and the guidance module. The decision module is active during both training and execution, it reconstructs the agent-wise state and feeds the agent-wise state to make decisions. The guidance module is only active during training and is used to guide the agent-wise state reconstruction. As shown in Figure 1 (b), the guidance module shifts the focus to the agents rather than the dimensional feature. By dynamically adjusting attention for each agent, potential relationships between agents are explicitly captured, avoiding overlap and improving the decision-making. Since both training and execution rely on the reconstructed agent-wise state for decision-making, as shown in Figure 1 (d), the inter-agent relationships are incorporated into the agent-wise state, and the additional error is reduced through state consistency, enhancing decision-making effectiveness. To evaluate the performance of RGP, we conduct extensive experiments on discrete and continuous environments, experimental results demonstrate the effectiveness.

In summary, our contributions are: i) We propose a novel method that maintains consistency between training and execution, reducing error accumulation. ii) Our method reconstructs a more comprehensive agent-wise state, which is extracted based on agents rather than the dimension-wise state, capturing inter-agent relationships and enhancing their performance. iii) Experimental results show that our method outperforms existing knowledge distillation-based approaches, especially in dynamic, complex environments.

## 2 PRELIMINARY

**Dec-POMDP:** A fully cooperative multi-agent reinforcement learning task comprising $n$ agents can be represented by a *Decentralized Partially Observable Markov Decision Process* (Dec-

POMDP) (Oliehoek et al., 2016), which is defined as $\langle \mathcal{N}, \mathcal{A}, \mathcal{S}, \mathcal{P}, \mathcal{O}, \Omega, r, \gamma \rangle$, where $\mathcal{N} = \{1, ..., n\}$ is a set of agents, $\mathcal{A} = \left\{ \mathcal{A}^i \right\}_{i \in \mathcal{N}}$ is the set of joint actions $\boldsymbol{a}_t = \left\{ a_t^i \right\}_{i \in \mathcal{N}}$ at time step $t$. $\mathcal{S}$ is a set of global states $\boldsymbol{s}_t$, which is computed by state transition function $\mathcal{P}(\boldsymbol{s}_{t+1}|\boldsymbol{s}_t, \boldsymbol{a}_t) : \mathcal{S} \times \mathcal{A} \times \mathcal{S} \mapsto [0, 1]$. Global state $\boldsymbol{s}_t$ is a high-dimensional vector which contains the information of all agents and environment at time $t$. For agent $i$, we denote the vector that indicates its inter-relationships with other units as the **agent-wise state** $\bar{s}_t^i$. $\mathcal{O}$ is a set of local observations $o_t^i$, which is generated by the joint observation function $\Omega(\boldsymbol{o}_{t+1}|\boldsymbol{a}_t, \boldsymbol{s}_{t+1}) : \mathcal{S} \times \mathcal{A} \mapsto \mathcal{O}$. All agents share the same reward function $r(\boldsymbol{s}_t, \boldsymbol{a}_t) : \mathcal{S} \times \mathcal{A} \mapsto (\mathbb{R})$ and $\gamma \in [0, 1)$ is the discount factor. Each agent has an action-observation history $\tau^i \in \mathcal{J} \equiv (\Omega \times \mathcal{A})^*$. Based on this history, the agent conditions a stochastic policy $\pi^i(a_t^i|\tau_{t-1}^i, o_t^i) : \mathcal{J} \times \mathcal{A} \mapsto [0, 1]$. The joint action-observation history $\boldsymbol{\tau} \in \mathcal{J}$ is defined similarly. Based on the observed history $\tau^i$, we introduced a **trajectory state** $\bar{h}_t^i$ for the agent, which is a high-dimensional vector represents the action-observation history of agent $i$. The joint policy $\boldsymbol{\pi}$ has a *joint action-value function*: $Q_{tot}^{\boldsymbol{\pi}}(\boldsymbol{s}_t, \boldsymbol{a}_t) = \mathbb{E}_{\boldsymbol{s}_{t+1:\infty}, \boldsymbol{a}_{t+1} \sim \boldsymbol{\pi}}[G_t|\boldsymbol{s}_t, \boldsymbol{a}_t]$, where $G_t = \sum_{k=0}^{T} \gamma^k r_{t+k}$. The ultimate goal is to obtain an *optimal joint policy* $\boldsymbol{\pi}^*$ with the *optimal joint action-value function* $Q_{tot}^{\boldsymbol{\pi}^*} = Q^*$.

**Diffusion Probabilistic Models:** Diffusion models (Ho et al., 2020; Rombach et al., 2022) generate samples matching the target data distribution by denoising Gaussian noise. The models consist of a forward process and a denoising process. The diffusion process introduces noise into the original distribution $p(\boldsymbol{x})$ until it becomes a pure Gaussian noise distribution, and can be described as a Markov process: $q(\boldsymbol{x}_k|\boldsymbol{x}_{k-1}) = \mathcal{N}(\boldsymbol{x}_k; \sqrt{1 - \beta_k}\boldsymbol{x}_{k-1}, \beta_k \boldsymbol{I})$, where $\boldsymbol{x}_0, ..., \boldsymbol{x}_K$ are latent variables, and $\boldsymbol{I}$ is an identity matrix. $\beta_k$ is the noise schedule measuring the proportion of noise added at each step. Given $\alpha_k = 1 - \beta_k$ and $\overline{\alpha}_k = \prod_{i=1}^{k}(1 - \beta_i)$, then $q(\boldsymbol{x}_k|\boldsymbol{x}_0) = \mathcal{N}(\boldsymbol{x}_k; \sqrt{\overline{\alpha}_k}\boldsymbol{x}_0, (1 - \overline{\alpha}_k)\boldsymbol{I})$. The denoising process starts with a Gaussian noise distribution $p(\boldsymbol{x}_K) = \mathcal{N}(\boldsymbol{0}, \boldsymbol{I})$ and restores the original data step by step, which can be described as: $p_\psi(\boldsymbol{x}_{k-1}|\boldsymbol{x}_k) = \mathcal{N}(\boldsymbol{x}_{k-1}; \mu_\psi(\boldsymbol{x}_k, k), \sum_\psi(\boldsymbol{x}_k, k))$, where $\mu_\psi(\boldsymbol{x}_k, k) = \frac{\sqrt{\alpha_k}(1 - \bar{\alpha}_k)}{1 - \bar{\alpha}_{k-1}}\boldsymbol{x}_k + \frac{\sqrt{\bar{\alpha}_{k-1}}\beta_k}{1 - \bar{\alpha}_k}\mathcal{G}_\psi(\boldsymbol{x}_k, k)$, and $\mathcal{G}$ presents a model used to reconstruct $\boldsymbol{x}$. It can be optimized by:

$$\mathcal{L}(\psi) = \mathbb{E}_{k \sim [1, K], \boldsymbol{x}_0 \sim q(\boldsymbol{x}_0), \zeta \sim \mathcal{N}(\boldsymbol{0}, \boldsymbol{I})} \parallel \zeta - \zeta_\psi(\boldsymbol{x}_k, k) \parallel^2 \tag{1}$$

## 3 RELATED WORK

Multi-agent reinforcement learning faces the partial observability issue. To address this, Centralized Training with Decentralized Execution (CTDE) uses the global state to optimize the network during centralized training (Claus & Boutilier, 1998) while relying only on local observations for decentralized execution (Tan, 1993). To further mitigate partial observability, some methods (Guan et al., 2022; Wang et al., 2021) adopt agent communication, which allows agents to exchange information with each other. However, such methods typically require expensive communication channels. Information-sharing methods provide additional information for training and gradually reduce the additional information for distributed execution, such as SUPER (Gerstgrasser et al., 2024), which shares prioritized experience; MACPF (Wang et al., 2022), which shares the actions of other agents; and CADP (Zhou et al., 2023), which shares suggestions between agents. But this can reduce the performance of the agent during the execution phase. Other methods adopt a modeling approach. Opponent modeling methods (Papoudakis et al., 2021; Sun et al., 2024) overlook the potential environmental information. World model methods (Liu et al., 2024) are employed to model the entire environment directly. However, modeling errors can be progressively amplified through interactions and feedback loops between agents, thereby affecting the overall learning performance and the convergence of strategies. State reconstruction methods aim to learn a model to reconstruct the global state during execution. For instance, PTDE (Chen et al., 2022) uses knowledge distillation to train a student network that reconstructs linear mapping of the global state. SIDiff (Xu et al., 2024) reconstructs the global state through diffusion. They have incorporated the inconsistent dimension-wise state for decision-making in training and execution, which introduces additional errors and suboptimal decisions.

## 4 METHOD

To capture the inter-agent relationships and decrease the additional errors, we propose a method called Reconstruction-Guided Policy (RGP), which consists of a decision module and a guidance

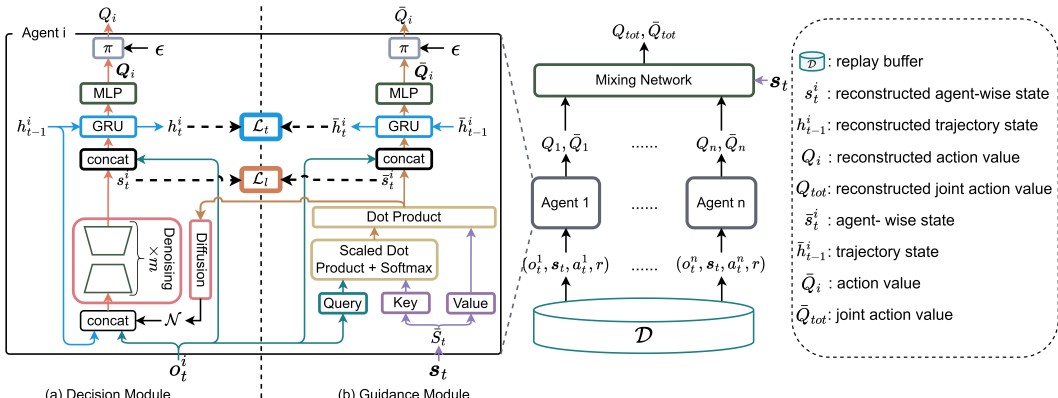

Figure 2: The framework of RGP. Figure (a) illustrates the Decision Module, which is active during both training and execution, reconstructing agent states $\bar{s}_t^i$ to assist in decision-making and action selection. Figure (b) shows the Guidance Module, which is only active during training, generating agent states $s_t^i$ as ground truth to guide the training of the generative model. Both modules are optimized through the same mixing network.

module. As illustrated in Figure 2, the decision module is designed to reconstruct the agent-wise state and make decisions, which is active in both training and execution. The guidance module is designed to guide the learning of the decision module, which is only active in the training. In the following, we first describe the decision module, followed by the guidance module.

## 4.1 THE DECISION MODULE

As we discussed previously, the dimension-wise state lacks inter-agent dependencies, which is unfavorable for decision-making. Therefore, the decision module takes the observation and trajectory to reconstruct the agent-wise state, and makes decisions based on the agent-wise state.

Specifically, at arbitrary step $t$ given the agent $i$'s observation $o_t^i$ and latest reconstructed trajectory state $h_{t-1}^i$, our decision module takes the following steps to make the decision $a_t^i$.

First, it takes the observation $o_t^i$ and latest reconstructed trajectory state $h_{t-1}^i$ as a condition to reconstruct the agent-wise state of agent $i$, which represents its inter- relationships with other units as we discussed previously. Considering the generative models (Kelkar & Anastasio, 2021; Qi et al., 2023; Chen et al., 2018)are effective for reconstruction, we apply a generative model to reconstruct the agent-wise state of the agent:

$$s_t^i = \mathcal{G}([o_t^i, h_{t-1}^i]), \tag{2}$$

where $[\cdot, \cdot]$ denotes the concatenation. $s_t^i$ denotes the reconstructed agent-wise state of agent $i$. $\mathcal{G}(\cdot)$ denotes the generative model and we implement the $\mathcal{G}(\cdot)$ with diffusion model, which reconstructs the agent-wise state by iteratively conducting Eq. (3) $K$ times to obtain $s_{t,0}^i$, and use $s_{t,0}^i$ as $s_t^i$:

$$s_{t,k-1}^i = \frac{s_{t,k}^i}{\sqrt{\alpha_k}} - \frac{\beta_k}{\sqrt{\alpha_k(1-\bar{\alpha}_k)}}\zeta_\psi(s_{t,k}^i, [o_t^i, h_{t-1}^i], k) + \sqrt{\beta_k}\zeta, \zeta \sim \mathcal{N}(\mathbf{0}, \boldsymbol{I}), \text{for } k = K, ..., 1. \tag{3}$$

$\beta_k$ is the noise schedule measuring the proportion of noise added at each step, and $\alpha_k = 1 - \beta_k$. It is worth noting that although we chose diffusion models as the generative model in this work, other generative models are also applicable (we compared the results of reconstructing the agent-wise state using different generative models, and the detailed results can be found in Appendix C.2.)

Then, the reconstructed agent-wise state is concatenated with the observation and fed to a function $f$ to predict the optimal action and the corresponding individual Q-value of the optimal action:

$$h_t^i, a_t^i, Q_i = f([s_t^i, o_t^i], h_{t-1}^i), \tag{4}$$

where function $f$ is composed of a recurrent neural network (RNN) (Zaremba, 2014; Graves & Graves, 2012; Chung et al., 2014) and an individual Q-network (Mnih, 2013). Particularly, the

concatenation of reconstructed agent-wise state $s_t^i$ and observation $o_t^i$ is first fed to a RNN to obtain the reconstructed trajectory state of the current step:

$$h_t^i = \text{RNN}([s_t^i, o_t^i], h_{t-1}^i), \tag{5}$$

where $\text{RNN}(\cdot)$ denotes the RNN, and we implemented with Gated Recurrent Unit (GRU)(Chung et al., 2014). $h_t^i, h_{t-1}^i$ denote the reconstructed trajectory state of the current step and previous step, and $h_0^t$ is initialized with a zero vector, that is $h_t^i = \mathbf{0}$. Then, the reconstructed trajectory state of the current step is fed to the individual Q-network, which is implemented with multiple-layer perceptron (MLP), to predict the individual Q-value of each action and obtain the action.

$$a_t^i = \epsilon\text{-Greedy}(\arg\max_a \text{MLP}(h_t^i, a)) , \tag{6}$$

$$Q_i = \text{MLP}(h_t^i, a_t^i) , \tag{7}$$

where $a_t^i$ is the action taken by agent $i$ to interact with the environment. $\epsilon$ is the hyperparameter in training, and $\epsilon = 0$ in execution, $Q_i$ denotes the individual Q-value.

It is obvious that the reconstruction of the agent-wise state is uncontrollable within the decision module, which might lead to bad decisions in Eq.(6). Therefore, we introduce a guidance module to guide the reconstruction.

## 4.2 THE GUIDANCE MODULE

The guidance module is a semi-siamese network paired with the decision module, guiding the reconstruction and the learning of the decision module. As shown in Figure 2 (b), the guidance module guides the decision module by incorporating the global state of the environment to estimate the agent-wise state $\bar{s}_t^i$ and trajectory state $\bar{h}_t^i$. They are derived from the true global state and can therefore act as constraints that guide the decision module to reconstruct the high-fidelity reconstructed agent-wise state and reconstructed trajectory state when the global state is unavailable. It is important that the primary function of the guidance module is to guide the learning of the decision module, therefore, it is active only during training. In the following, we will first introduce the pipeline of the guidance module and then explain how it guides the decision module.

**The Pipeline of the Guidance Module.**   At arbitrary step $t$ given the agent $i$'s observation $o_t^i$ and the global state $s_t$, our guidance module works with the following steps and produces the guidance agent-wise state $\bar{s}_t^i$ and guidance trajectory state $\bar{h}_t^i$ (Here, we prefix the variables produced by the guidance module with "guidance" to avoid confusion).

Since $s_t$ is a vector that contains the state of all the agents in the environment (if environmental information is present, we treat it as an independent agent), we first decompose the global state by agent, mapping each agent's state to the same dimension $d$, stacking them into a matrix $S_t$. $S_t \in \mathbf{R}^{n \times d}$, each row of the matrix $S_t$ denotes the state of one agent. Then we encode the ID of the agent, and concatenate them with global state matrix and observation. That is:

$$\bar{S}_t = [S, P], \bar{o}_t^i = [o_t^i, p^i], , \tag{8}$$

where $P$ is the embedding matrix of agents' ID, which is a learnable matrix and each row vector of which corresponds to one agent. $p^i$ is the $i$-th row vector of $P$, which denotes the embedding of agent $i$'s ID. Then, we apply a multi-head attention on $\bar{S}_t$ and $\bar{o}_t^i$ to compute the guidance agent-wise state, which represents the inter- relationships with other units:

$$\bar{s}_t^i = \text{Attn}(f_q(\bar{o}_t^i), f_k(\bar{S}_t) , f_v(\bar{S}_t)), \tag{9}$$

where $f_q, f_k, f_v$ are implemented with MLP. $\text{Att}(\cdot)$ denotes the multi-head attention, in which $f_q(\bar{o}_t^i), f_k(\bar{S}_t), f_v(\bar{S}_t)$ act as the query, key and value correspondingly. $\bar{s}_t^i$ denotes the guidance agent-wise state of agent $i$. Next, we feed the guidance agent-wise state $\bar{s}_t^i$ to $f$ (Eq.4) to obtain guidance trajectory state, guidance action and guidance Q-value:

$$\bar{h}_t^i, \bar{a}_t^i, \bar{Q}_i = f([\bar{s}_t^i, o_t^i]) . \tag{10}$$

**The Guidance on Decision Module.** To guide the reconstruction of the agent-wise state, we treat $\bar{s}_t^i$ as the ground truth data that diffusion model $\mathcal{G}$ is supposed to reconstruct given the latest trajectory state and observation. Therefore, we apply the learning target of the diffusion model to guide the agent-wise state reconstruction in decision module.

$$\mathcal{L}_l = \mathbb{E}_{k \sim [1,K], \bar{s}_0^i \sim q(\bar{s}_0^i), \zeta \sim \mathcal{N}(\mathbf{0},\mathbf{I})} \parallel \zeta - \zeta_\psi(\sqrt{\bar{\alpha}_k}\bar{s}_t^i + \sqrt{1 - \bar{\alpha}_k}\zeta, [o_t^i, \bar{h}_{t-1}^i], m) \parallel^2, \quad (11)$$

Recall that one of the factors influencing the decision made by the decision module is the trajectory state ($h_{t-1}^i$ in Eq.(4)), which is computed from the agent-wise states of previous steps $s_{t-1}^i$. As decisions are made sequentially over multiple time steps, any slight inaccuracy in the reconstructed agent-wise state at each step can accumulate in the trajectory state, affecting subsequent decisions. To address this, we utilize the guidance trajectory state generated by the guidance module to refine the reconstruction of the agent-wise state in an indirect manner. Specifically, the guidance trajectory state is employed to constrain the trajectory state derived from the reconstructed agent-wise state:

$$\mathcal{L}_t = \mathbb{E}_{\mathcal{D}}[(\bar{h}_t^i - h_t^i)^2]. \quad (12)$$

### 4.3 MODEL LEARNING

Although the guidance module guides the agent-wise state reconstruction of the decision module, the parameters of both the decision and guidance modules are randomly initialized. The constraints from Eq.(11) and Eq.(12) can only make the local and trajectory states generated by the two modules closely approximate each other, but cannot guide them toward convergence in a direction favorable for making optimal decisions. Following previous work, we also adopt a Mixing network (Rashid et al., 2020) to further constrain the agent's decisions. Specifically, we apply two TD losses (Peng & Williams, 1993; Mnih et al., 2015) to constrain the actions produced by the decision module and the guidance module.

$$\mathcal{L}_d = \mathbb{E}_{\mathcal{D}}[(r + \gamma \max_{\boldsymbol{a}_{t+1}} Q_{tot}(\boldsymbol{\tau}_{t+1}, \boldsymbol{a}_{t+1}, \boldsymbol{s}_{t+1}; \theta^-, \psi^-) - Q_{tot}(\boldsymbol{\tau}_t, \boldsymbol{a}_t, \boldsymbol{s}_t; \theta, \psi))^2], \quad (13)$$

$$\mathcal{L}_g = \mathbb{E}_{\mathcal{D}}[(r + \gamma \max_{\bar{\boldsymbol{a}}_{t+1}} \bar{Q}_{tot}(\bar{\boldsymbol{\tau}}_{t+1}, \bar{\boldsymbol{a}}_{t+1}, \boldsymbol{s}_{t+1}; \theta^-, \eta^-) - \bar{Q}_{tot}(\bar{\boldsymbol{\tau}}_t, \bar{\boldsymbol{a}}_t, \boldsymbol{s}_t; \theta, \eta))^2], \quad (14)$$

where $\psi$ are the parameters of the generative model, $\eta$ are the parameters of the attention and $\theta$ are the parameters of the other networks. $\psi^-$, $\eta^-$ and $\theta^-$ represent the parameters of the target network, which are periodically copied the parameters and kept constant for a definite number of iterations. It is important to note that although the actions generated by the guidance module are not used for interaction with the environment, constraining them helps guide the agent-wise state and trajectory states to converge toward optimal actions.

Therefore, the learning target of our method is:

$$\mathcal{L} = \mathcal{L}_l + \mathcal{L}_t + \mathcal{L}_d + \mathcal{L}_g. \quad (15)$$

The details of training are illustrated in Appendix A.1, algorithm1.

## 5 EXPERIMENTS

We conduct extensive experiments to evaluate the performance of RGP, and we particularly focus on the research questions: i) How does RGP perform compared with other methods (**RQ1**)? ii) Can RGP reduce the gap between training and execution (**RQ2**)? iii) Why does RGP can achieve better performance than other methods(**RQ3**)? iv) Can RGP explore the potential relationships between agents? (**RQ4**)? v) Can RGP adapt to continuous action environments? (**RQ5**)? vi) How does the RGP perform under more challenge partially observable conditions? (**RQ6**)?

### 5.1 EXPERIMENT SETTING

**Environments.** We primarily evaluated RGP on SMAC (Samvelyan et al., 2019) and SMACv2 (Ellis et al., 2024). SMAC is the most widely used discrete multi-agent environment, while SMACv2 introduces stochasticity based on SAMC. We set up the SMACv2 maps with 5 ally agents against 5 enemies. Additionally, to further demonstrate the portability of RGP, we conducted experiments in continuous predator-prey and continuous cooperative navigation scenarios (Lowe et al., 2017). For detail of continuous predator-prey and continuous cooperative navigatio, see Appendix B.

| MAP | VDN | QMIX | QPLEX | CADP | PTDE | SIDiff | RGP | HPN-QMIX | RGP+HPN |
|---|---|---|---|---|---|---|---|---|---|
| 27m_vs_30m | 92.5±6.0 | 96.4±3.5 | 96.7±3.0 | 92.6±4.5 | 96.6±1.9 | 93.2±3.9 | **98.1±2.1** | 99.5±0.7 | **100.0±0.0** |
| MMM2 | 88.3±5.4 | 92.7±5.4 | 94.8±2.9 | 92.0±4.2 | 94.7±3.4 | 87.2±8.4 | **95.1±3.5** | 99.4±0.3 | **100.0±0.0** |
| 3s5z_vs_3s6z | 65.9±5.4 | 70.8±6.4 | 30.6±37.3 | **90.2±3.1** | 67.4±6.9 | 10.0±8.7 | 76.9±2.8 | 94.7±2.2 | **96.6±2.3** |
| corridor | 86.9±6.9 | 92.1±4.6 | 30.5±39.9 | 87.8±4.3 | 91.4±2.7 | 70.7±17.0 | **93.3±3.0** | 96.4±3.4 | **96.7±2.9** |
| 6h_vs_8z | 23.3±17.9 | 38.1±28.3 | 7.6±2.3 | 51.8±16.3 | 52.4±8.5 | 23.8±15.9 | **73.1±4.6** | 94.4±3.5 | **95.6±2.3** |
| Protoss_360 | 58.8±5.2 | 69.4±4.6 | 70.0±5.3 | 64.6±6.8 | 64.4±3.4 | 25.3±11.4 | **76.2±3.6** | 79.4±4.8 | **84.5±4.3** |
| Terran_360 | 65.1±6.6 | 72.8±5.9 | 74.6±6.5 | 70.2±5.9 | 63.4±6.3 | 47.8±10.9 | **75.0±5.9** | 78.0±5.2 | **82.8±4.4** |
| Zerg_360 | 51.6±7.1 | 55.6±5.3 | 40.9±28.7 | 56.7±7.7 | 55.3±5.6 | 46.6±8.9 | **61.4±7.1** | 67.4±7.0 | **70.1±7.0** |

Table 1: The mean win rate (%) of different methods on SMAC and SMACv2, with ± denoting the standard deviation. We applied RGP to QMIX, denoted as **RGP**. Protoss_360 indicates that in the protoss map, agents have a 360° field of view. Similarly, zerg_360 and terran_360 represent analogous settings for the zerg and terran maps. We further applied RGP to HPN-QMIX, denoted as GRP+HPN. The corresponding training plots are provided in Appendix C.1 for reference.

**Baselines.** In discrete action environments, we used traditional CTDE methods as baselines, including VDN (Sunehag et al., 2017), QMIX (Rashid et al., 2020), QPLEX (Wang et al., 2020), and HPN-QMIX (Jianye et al., 2022). Additionally, we compared RGP with CADP (Zhou et al., 2023), which enhances CTDE by providing agents with supplementary information during execution. We further evaluated our method against PTDE (Chen et al., 2022) and SIDiff (Xu et al., 2024), which reconstruct the dimension-wise state through local observations during decentralized execution. In continuous action environments, we compared RGP with policy-based methods, such as MADDPG (Lowe et al., 2017) and FACMAC (Peng et al., 2021).

**Implementation Details.** Our model was trained on a setup with 4 NVIDIA A40 GPUs, an Intel Gold 5220 CPU, and 504GB of memory, optimized using the Adam optimizer (Kingma & Ba, 2014). Due to limited computational resources, we replaced the Unet used in the original paper DDPM (Ho et al., 2020; Rombach et al., 2022) with an MLP. We set the timestep of diffusion to 10, and the heads of attention to 4. The details of other hyperparameters can be found in Appendix A.2 table 4. We also referred to PyMARL2 (Hu et al., 2021) for extensive tuning and proper training of all baselines to ensure they achieved optimal performance.

## 5.2 BENCHMARK RESULTS

We compare RGP to baseline methods with respect to the normalized battle mean win rate obtained during online evaluation. We conducted 3 trials with different seeds, reporting the average results and standard deviation. The results of RGP and baseline methods are summarized in Table 1.

From Table 1, we can observe that (RQ1): (1) Compared with the methods without reconstructing the states, such as VDN, QMIX, QPLEX, our RGP achieves the best performance, which demonstrates that relying solely on local observations is insufficient in distributed execution. Additional information is needed to assist the agent in decision-making to achieve better performance. (2) We noticed that RGP outperformed QMIX in most cases, which has exactly the same backbone with QMIX. Similarly, RGP+HPN also outperformed HPN-QMIX. That demonstrates that the reconstructed agent-wise state provided by the decision module beneficial for the decision of agent. (3) Compared with the methods that reconstruct the states, such as PTDE and SIDiff, our RGP achieves the best performance in most cases, which shows that our method provides agents with more refined additional information, specifically the agent-wise state, through the decision and guidance modules, which can effectively enhance the agents' decision-making performance.

## 5.3 PERFORMANCE RETENTION

To investigate whether RGP reduces the performance gap between training and execution (RQ2), we compare the retention ratio of RGP with the PTDE and SIDiff, which also reconstruct the state for decision making, on the environment of SMAC and SMACv2, and illustrate the results in Table 2. From Table 2, we can observe that: PTDE and SIDiff exhibit significant performance discrepancies between centralized training and distributed execution, with both methods having a lower performance retention rate than our proposed RGP. Since both PTDE and SIDiff aim to reconstruct the dimension-wise state and have an inconsistent state in decision-making of training and execution,

| Map | PTDE | | | SIDiff | | | RGP | | |
|---|---|---|---|---|---|---|---|---|---|
| | CE | DE | PRR | CE | DE | PRR | CE | DE | PRR |
| 3s5z_vs_3s6z | 77.6±6.2 | 67.4±6.9 | 86.9±11.3 | 10.5±2.8 | 10.0±8.7 | 95.2±8.7 | 79.3±2.1 | 76.9±2.8 | **97.0±4.4** |
| 6h_vs_8z | 71.2±5.3 | 52.4±8.5 | 73.6±13.1 | 29.7±22.8 | 23.8±15.9 | 80.1±81.6 | 76.5±6.1 | 73.1±4.6 | **95.6±9.7** |
| protoss_360 | 73.2±5.5 | 64.4±3.4 | 88.0±8.1 | 59.8±5.7 | 25.3±11.4 | 42.3±19.5 | 76.9±4.1 | 76.2±3.6 | **99.1±7.1** |
| terran_360 | 71.6±4.8 | 63.4±6.3 | 88.5±10.6 | 68.1±6.5 | 47.8±10.9 | 70.2±17.4 | 73.1±3.8 | 75.0±5.9 | **102.6±9.7** |

Table 2: The mean win rates (%) and performance retention ratios (%) (Chen et al., 2022) of different methods on SMAC. We tested their results under both centralized and decentralized execution. CE stands for centralized execution, DE stands for decentralized execution. PRR stands for Performance Retention Rate, which can be calculated by dividing the win rates of DE by it of CE. The uncertainty of PRR is measured by the standard deviation propagation technique.

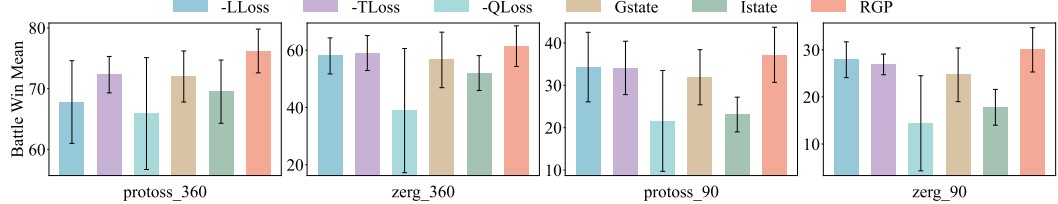

Figure 3: The ablation study. We tested the mean win rate under different variants of RGP. Protoss_90 indicates that the agents have a 90° field of view in the protoss map, and zerg_90 represents a similar setting for the zerg map.

this result suggests that learning the reconstruction of the agent-wise state and adopting a consistent state in decision-making is beneficial.

## 5.4 ABLATION STUDY

To further investigate the factors that support the performance of RGP (RQ3), we conduct the ablation study. specifically, we compare the performance of RGP with the following variants:

- **-LLoss** removes the constraint ($\mathcal{L}_l$, Eq. 11) in the guidance module. That means the reconstruction of the agent-wise state is only guided by the constraint on trajectory state ($\mathcal{L}_t$).

- **-TLoss** removes the constraint on the trajectory state ($\mathcal{L}_t$, Eq.12) in the guidance module. That means the reconstruction of the agent-wise state is only guided by the constraint on agent-wise state ($\mathcal{L}_l$).

- **-QLoss** removes the constraint ($\mathcal{L}_g$, Eq.14) on the Q-value.

- **Gstate** applies the global state reconstruction in decision module and make decision based on the the reconstructed global state.

- **Istate** applies the inconsistent state in training and execution on RGP.

The results are illustrated in Figure 3, from which we have following findings: (1) Removing the constraint on the Q-value of guidance module (-QLoss) significantly decreased the performance of RGP and introduced greater variance. This is because without the constraint of $\mathcal{L}_g$, the guidance module struggles to guide the reconstructed agent-wise state and trajectory state to optimal-favorable states, hindering the agent's ability to make optimal decisions. (2) Removing $\mathcal{L}_l$ (-LLoss) and $\mathcal{L}_t$ (-TLoss) decrease the performance. This is because both $\mathcal{L}_l$ and $\mathcal{L}_t$ is used to guide the agent-wise state reconstruction. Removing one of them affects the reconstruction of the agent-wise state, thereby impacting overall performance. Also, we observed that the impact of both settings on performance is less significant than that of -QLoss. This is because removing one constraint still leaves the other constraint in place to properly guide the reconstruction of the agent-wise state. (3) Making decisions based on the reconstructed global state (GState) instead of the agent-wise state decreases the performance. We believe this is due to two facts. On one hand, making decisions directly based on the global state without considering the agent-wise states of the agents results in decisions that do not align with the actual circumstances of the agents. On the other hand, the global state itself is discrete, and the diffusion models we implemented may not be adept at generating discrete

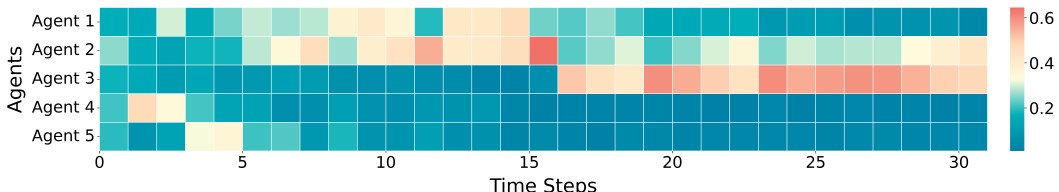

Figure 4: The attention weight of each agent.

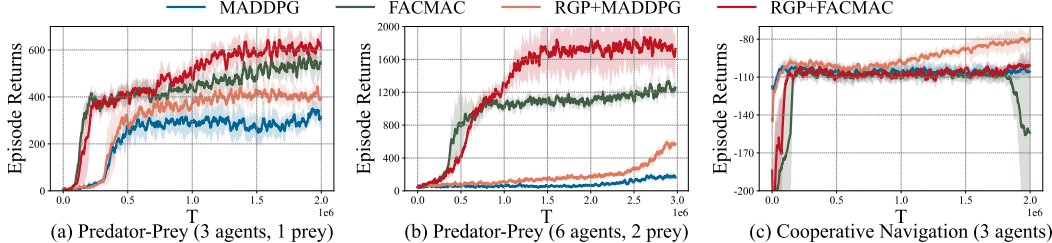

(a) Predator-Prey (3 agents, 1 prey)    (b) Predator-Prey (6 agents, 2 prey)    (c) Cooperative Navigation (3 agents)

Figure 5: The average reward of different methods. **RGP+MADDPG** denotes using MADDPG as the subsequent information processing and critic network, and **RGP+FACMAC** refers to using FACMAC as the critic network. The experiment is conducted using three random seeds. The thick, dark-colored lines represent the mean, while the shaded area indicates the standard deviation.

data, leading to inherent biases in the reconstructed global state. 4) Applying inconsistent states to make decisions in training and execution (IState) decreases the performance. The reason as we discussed previously, the policy is trained on the true state, never trained to handle imperfect states (the reconstructed one), which might be slightly different from the actual one due to reconstruction inaccuracies. Feeding the imperfect states to policy increase the suboptimal action.

## 5.5 AGENT-WISE STATE FOR AGENT

To further verify whether our RGP indeed accounts for the potential relationships between agents (RQ4), we randomly select an arbitrary agent and visualize the weight of attention assigned to other agents when computing the agent-wise state for decision-making. The results are illustrated in Figure 4, in which the agent is denoted in Agent 1, and other agents are denoted with Agent $2-5$. The horizontal axis represents time steps. Each cell indicates the attention weight that Agent 1 assigns to other agents when calculating the agent-wise state. The redder the color, the higher the weight of the corresponding agent's state in Agent 1's agent-wise state.

From Figure 4, we observed that: (1) Before step 15, Agent 1 focuses on its own state, but after step 15, its reliance on its own state decreases. This is likely because, in the early stages, it needs its own state to position itself in the environment, but once positioning is complete, it relies only on local observations and no longer needs its own state. (2) Before step 15, the weight of Agent 2 is relatively high, while after step 15, the weight of Agent 3 increases more than that of Agent 2. We assume that it is likely because, Agent 1 dynamically adjusts its cooperation strategy based on the evolving importance of other agents, first relying more on Agent 2 and later shifting its focus to Agent 3 as the context changes. This suggests that the system is able to flexibly respond to varying circumstances, optimizing collaboration as needed throughout the task. These results demonstrate that RGP effectively captures inter-agent relationships and selects information of important agents.

## 5.6 PORTABILITY ON CONTINUOUS ACTION ENVIRONMENT

To further investigate the portability of RGP on continuous action environment (RQ5), we transport the decision module and the guidance module of RGP to the MADDPG (Lowe et al., 2017) and FACMAC (Peng et al., 2021), which tackle continuous action scenarios, and compare their performance in the continuous action environment: Cooperative Navigation and Predator-Prey (Lowe et al., 2017). The results are illustrated in Figure 5, in which RGP+MADDPG denotes the setting we transport the RGP to MADDPG, and RGP+FACMAC denotes the setting we transport RGP to FACMAC (for the details of implementation please refer Appendix A.1, Algorithm 2).

| Metric | protoss_360 | protoss_90 | protoss_30 | terran_360 | terran_90 | terran_30 | zerg_360 | zerg_90 | zerg_30 |
|---|---|---|---|---|---|---|---|---|---|
| QMIX | 69.4±4.6 | 27.6±8.2 | 7.2±1.6 | 72.8±5.9 | 41.9±5.3 | 5.2±2.0 | 55.6±5.3 | 23.2±5.6 | 2.1±0.6 |
| RGP | 76.2±3.6 | 37.2±6.5 | 10.1±5.0 | 75.0±5.9 | 46.3±4.1 | 9.4±1.2 | 61.4±7.1 | 30.0±4.7 | 4.2±1.1 |
| PIR | 9.8±5.2 | 34.8±25.7 | 40.3±70.0 | 3.0±8.1 | 10.5±9.8 | 80.8±38.7 | 10.4±12.8 | 29.3±21.5 | 100.0±59.7 |
| HPN_QMIX | 79.4±4.8 | 47.5±5.8 | 8.8±3.6 | 78.0±5.2 | 49.5±7.4 | 4.4±2.8 | 67.4±7.0 | 17.1±5.3 | 2.5±2.3 |
| RGP+HPN | 84.5±4.3 | 53.9±6.5 | 10.2±3.6 | 82.8±4.4 | 60.0±5.6 | 12.8±3.4 | 70.1±7.0 | 26.3±8.0 | 4.1±1.6 |
| PIR | 6.4±5.4 | 13.5±13.8 | 15.9±41.4 | 6.2±5.7 | 21.2±11.8 | 190.9±144.0 | 4.0±10.4 | 53.8±49.7 | 64.0±87.0 |

Table 3: The winning rates (%) and performance improving ratios (PIR)(%) of different methods on SMACv2. We tested their results under decentralized execution and calculated the performance improving ratio during decentralized execution. Protoss_30 indicates that the agents have a 30° field of view. Similarly, zerg_30 and terran_30 represent analogous settings for the zerg and terran maps, respectively. The uncertainty of PIR is measured by the standard deviation propagation technique.

We can observe from Figure 5 that: (1) RGP+MADDPG achieves better performance than MAD-DPG, and RGP+FACMAC achieves better performance than FACMAC in all the settings. That shows introducing our RGP benefits the performance, and our RGP is not only applicable to discrete action environments but also to continuous action environments. (2) In the Predator-Prey (Figure 5 (a) and (b)), RGP+FACMAC achieves the average return which several times than FACMAC. Especially in the setting with 6 agents and 2 prey, the average reward of FACMAC is around 1200, while the average reward of RGP+FACMAC is around 1600! We assume that is because the introduction of agent-wise states helps agents effectively cooperate with surrounding agents, facilitating their collaboration to better corner the prey.

## 5.7 Further Investigation

In multi-agent decision-making, the field of view of an agent is a key factor affecting its decisions; a smaller field of view makes it more likely for the agent to make suboptimal decisions. To investigate the performance of RGP under more challenging partially observable conditions (RQ6), we tested RGP's performance across different fields of view (360°, 90°, and 30°) and compared it with QMIX and HPN-QMIX under the same settings.

The results are illustrated in Table 3, in which both RGP+HPN and RGP incorporate the decision module and the guidance module, whereas HPN-QMIX and QMIX lack these modules. Moreover, QMIX is the backbone of RGP, and HPN-QMIX is the backbone of RGP+HPN. From Table 3, we can observe that: (1) Regardless of the field of view, RGP outperforms the QMIX and RGP+HPN outperforms the HPN-QMIX, which demonstrates the effectiveness of our method again. (2) In all settings, as the field of view decreases, the difference between the setting of introducing the decision module and guidance module and the setting without these two modules becomes increasingly pronounced. It suggests that our method can reconstruct a state that better characterizes the actual state of the agent for decision-making.

## 6 Conclusion

In this paper, we propose a method called Reconstruction-Guided Policy (RGP), which consists of a decision module and a guidance module. The decision module is responsible for reconstructing the agent-wise state and making decisions, while the guidance module helps guide the agent-wise state reconstruction. We conduct extensive experiments to evaluate the performance of RGP, and the results demonstrate the effectiveness of the proposed method. We found that ensuring consistency between training and execution states effectively prevents error amplification and improves performance retention ratios during execution, particularly in dynamic and complex environments. Additionally, considering the potential relationships between agents helps them identify interaction targets, which is beneficial for promoting cooperation among agents.

However, our method still has limitations. In environments where complex information such as images is used for state representation, it becomes difficult to decompose agent-specific information. In the future, techniques like object detection or image segmentation could be introduced to address this issue.

ACKNOWLEDGMENTS

The Jilin University team is partially supported by the Ministry of Science and Technology of China (No.2023YFF0905400) and the National Natural Science Foundation of China (No.62307020 and No.U2341229). The Shanghai Jiao Tong University team is partially supported by National Natural Science Foundation of China (62322603) and Shanghai Municipal Science and Technology Major Project (2021SHZDZX0102).

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

## A ALGORITHMS AND HYPERPARAMETERS

### A.1 RGP WITH VALUE DECOMPOSITION METHOD AND POLICY GRADIENT METHODS

Our method can be effectively integrated with both value decomposition and policy gradient approaches, which we have summarized as Algorithm 1 and Algorithm 2, respectively.

### A.2 HYPERPARAMETERS DETAIL

Details of RGP's hyperparameters are provided in Table 4. For baseline VDN, QMIX, QPLEX, they were implemented with the hyperparameters of PyMARL2 (Hu et al., 2021). For HPN-QMIX, CADP, PTDE, and SIDiff, they were implemented with their optimal hyperparameters, as specified in their respective papers (Jianye et al., 2022; Zhou et al., 2023; Chen et al., 2022; Xu et al., 2024).

## B ENVIRONMENT DETAILS

### B.1 CONTINUOUS PREDATOR-PREY

We followed FACMAC (Peng et al., 2021) and improved the simple tag scenario in MPE (Lowe et al., 2017), which is a variant of the classic predator-prey problem. Each agent operates in a continuous two-dimensional action space and must capture faster-moving prey in a randomly generated 2D environment, where obstacles hinder movement. To create a fully cooperative environment, the

---
**Algorithm 1** Training of RGP with Value decomposition methods

---
1: Initialize Generative model and attention
2: Initialize individual Q networks and target networks
3: Initialize mixing network and target mixing network
4: Initialize replay buffer $\mathcal{D}$
5: **for** each episode **do**
6:     Initialize global state $s_0$, guide hidden states and reconstruction hidden states
7:     **for** each timestep $t = 0$ to $T$ **do**
8:         **for** each agent $i = 1$ to $N$ **do**
9:             Obtain the global state $\boldsymbol{s}_0$ and local observation $o_t^i$
10:            Calculate $\bar{\boldsymbol{S}}_t$ and $\bar{o}_t^i$ according Eq.8
11:            Calculate agent-wise state $\bar{s}_t^i$ according Eq.9
12:            Calculate reconstructed agent-wise state $s_t^i$ according Eq.2
13:            Select action $a_t^i$ according to $s_t^i$ with $\epsilon$-greedy policy w.r.t $Q_i(o_i^t, s_t^i, h_i^{t-1})$
14:         **end for**
15:     Take joint action $\boldsymbol{a}_t$
16:     Obtain the global reward $r_{t+1}$, the next observation $\boldsymbol{o}_t$ and the next global state $\boldsymbol{s}_t$
17:     Store the episode in replay buffer $\mathcal{D}$
18:     Sample a batch of episodes from replay buffer $\mathcal{D}$
19:     Update the parameters of the decision module, guidance module and the mixing network according Eq. 15
20:     Replace target parameters every M episodes
21:     **end for**
22: **end for**

---

---
**Algorithm 2** Training of RGP with policy gradient methods

---
1: Initialize Generative model and attention
2: Initialize individual actor and the critic
3: Initialize replay buffer $\mathcal{D}$
4: **for** each episode **do**
5:     Initialize global state $s_0$, guide hidden states and reconstruction hidden states
6:     **for** each timestep $t = 0$ to $T$ **do**
7:         **for** each agent $i = 1$ to $N$ **do**
8:            Obtain the global state $\boldsymbol{s}_0$ and local observation $o_t^i$
9:            Calculate $\bar{\boldsymbol{S}}_t$ and $\bar{o}_t^i$ according Eq.8
10:            Calculate agent-wise state $\bar{s}_t^i$ according Eq.9
11:            Calculate reconstructed agent-wise state $s_t^i$ according Eq.2
12:            Select action $a_t^i$ according to $s_t^i$ with distribution w.r.t $\pi_i(o_i^t, s_t^i, h_i^{t-1})$
13:         **end for**
14:     Take joint action $\boldsymbol{a}_t$
15:     Obtain the global reward $r_{t+1}$, the next observation $\boldsymbol{o}_t$ and the next global state $\boldsymbol{s}_t$
16:     Store the episode in replay buffer $\mathcal{D}$
17:     **for** each agent $i = 1$ to $N$ **do**
18:         Sample a batch of episodes from replay buffer $\mathcal{D}$
19:         Fit value function by regression on mean-squared error and update critic
20:         Update actor according to policy gradient methods
21:     **end for**
22:     Update target parameters by polyak averaging
23:     **end for**
24: **end for**

---

| Module | Hyperparameter | Value |
|---|---|---|
| Decision Module | Diffusion process timestep | 10 |
| | Type of optimizer | Adam |
| | Learning rate | 0.001 |
| | Batch size | 128, (64 for 3s5z_vs_3s6z) |
| | TD lambda | 0.6, (0.3 for 6h_vs_8z) |
| | Training epochs | 10M, (8M for SMACv2) |
| | Buffer size | 5000, (2500 for 3s5z_vs_3s6z) |
| | Target update interval | 200 |
| Guidance Module | Attention heads | 4 |
| | Attention embedding dim | 32 |
| | Agent information mapping dim | 32 |

Table 4: Hyperparameter settings for the RGP training.

prey's strategy was designed as a hard-coded heuristic, where it always moves to the farthest position from the nearest predator at any given time. Agents can only receive rewards when they are close enough to the prey.

To introduce partial observability, a vision radius was added for each agent. Agents can only receive information within their vision radius (including other agents, prey, and obstacles), limiting their ability to observe other agents to approximately 60% of the time.

### B.2 CONTINUOUS COOPERATIVE NAVIGATION

The continuous cooperative navigation is a fully cooperative environment based on MPE (Lowe et al., 2017). In a two-dimensional space, $n$ agents must collaborate to cover landmarks scattered in the environment. Agents receive rewards from the environment when they get close enough to a landmark. They can only access information within their field of view, including their own position and velocity, the position and velocity of other agents, and the location of the landmarks. We implement the continuous action space by setting `continuous_actions=True` in the code. For more details, Further details are available in the official PettingZoo documentation [1].

## C  MORE EXPERIMENT

### C.1 TRAINING CURVES

To clearly illustrate the training procedure of the model, we present our curves in Figure 6, in which the x-axis represents the training episodes, while the y-axis represents the battle win rate in executing. It can be found that RGP+HPN outperforms HPN-QMIX and performs optimally in most cases. RGP also outperformed the rest of the baselines in most cases.

### C.2 THE IMPACT OF GENERATIVE MODELS

Our original intent of applying diffusion is to generate the agent-wise state. Therefore, diffusion serves as one module in our method. Theoretically speaking, any generative model, including VAE, GAN, or even simple MLP, can replace the diffusion in our method. To further validate the impact of generative models, we replaced the diffusion model in our RGP with MLP, VAE, and GAN respectively and conducted experiments in the SMACv2 environment. Experimental results are shown in Figure 7. We found that (1) Even though the diffusion model is replaced with VAE and MLP, we can observe that RGP outperforms the baseline QMIX in both settings, demonstrating the effectiveness of RGP. (2) Replacing the diffusion model with GAN, the performance slightly decreases. We assume that because GAN requires simultaneous training of both the generator and discriminator, and this stepwise training negatively affects the overall quality of generation. (3) We can observe that replacing the diffusion model with VAE and MLP results in performance much

---

[1] https://pettingzoo.farama.org/environments/mpe/

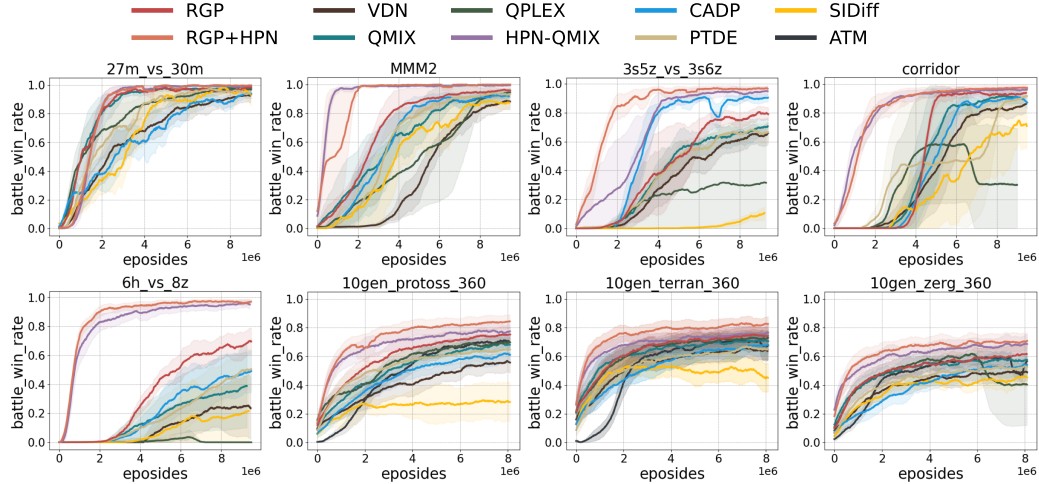

Figure 6: Training curves in SMACv2.

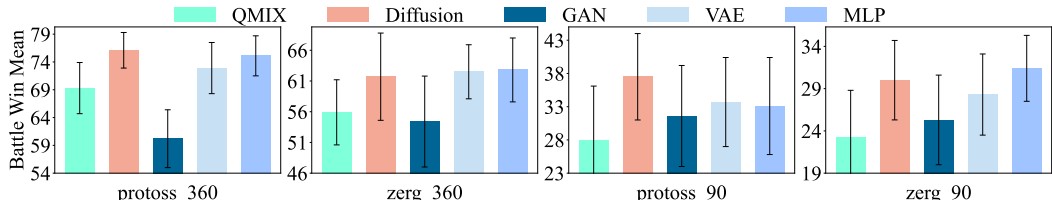

Figure 7: Performance comparison across various generative models. Protoss_360 and Protoss_90 represent agents having 360° and 90° fields of view, respectively, in the protoss map. Similarly, zerg_360 and zerg_90 represent analogous settings, respectively, in the zerg map.

closer to that of the diffusion setting. We assume the underlying reason is the substitution of the U-Net used in the original paper DDPM (Ho et al., 2020; Rombach et al., 2022) with MLP (to save the cost of computational resources). Nevertheless, the setting of diffusion still outperforms the setting of VAE and MLP in most cases.

## C.3 THE BIAS OF GENERATIVE MODELS

To explore the potential biases introduced by the generative model in state reconstruction, we conducted experiments on zerg_90. Specifically, we sampled 1,000 trajectories from PTDE and RGP, respectively, to calculate the cosine similarity between the true ground truth and the reconstruction. The results are visualized in Figure 8, in which each grid represents the cosine similarity for the corresponding agent at a specific time step. We can observe that (1) the reconstruction of our method is significantly closer to the true ground truth compared to the baseline. Furthermore, the similarity between the reconstruction of our method and the true ground truth consistently exceeds 0.9, demonstrating that our method introduces minimal bias during reconstruction. (2) The reconstruction performance of RGP in the first step is relatively poor compared to the subsequent steps. It is likely because of the cold state, in which the length of interaction history is zero, interferes with the reconstruction. However, even under this condition, the cosine similarity of RGP still remains above 0.9, indicating that this interference does not significantly impact the overall performance.

## C.4 MORE COMPLEX ENVIRONMENTS

### C.4.1 MORE AGENTS

To further explore RGP's performance in scenarios with a larger number of agents, we conducted experiments in the zerg_90_20_vs_20 environment. This setting involves 20 agents battling against

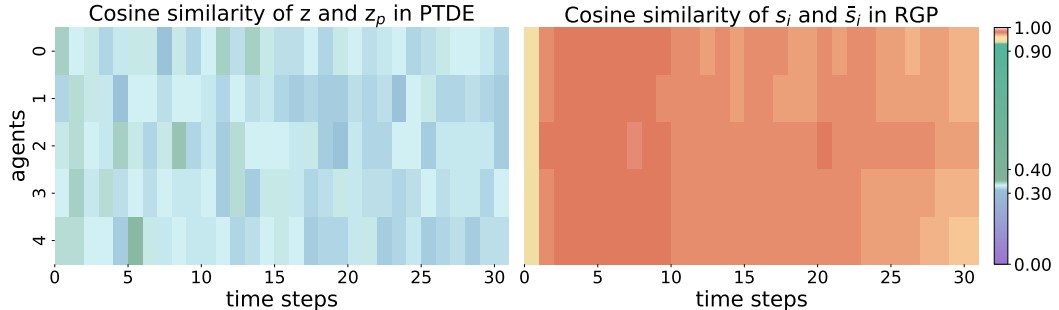

Figure 8: Cosine Similarity of PTDE and RGP. The heatmap's horizontal axis represents the timesteps, while the vertical axis corresponds to the agents. The low-dimensional representation of the true global information constructed by PTDE is denoted as $z$, and its reconstructed global information is denoted as $z_p$. To illustrate the detailed variations more clearly, we enhanced the color band by adding more vibrant colors to the key intervals require emphasis.

20 enemies under the zerg_90 configuration, presenting a significantly higher challenge for the algorithm. The experimental results are presented in Figure 9 (a). We observe that RGP outperforms all baselines, demonstrating its scalability in environments with a larger number of agents.

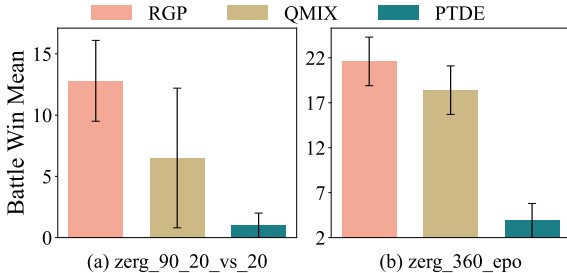

Figure 9: The winning rates (%) of different methods on more complex environments.

|  | RGP | QMIX |
|---|---|---|
| zerg | **7.4±4.5** | 2.7±1.9 |
| protoss | **14.2±4.2** | 1.6±0.1 |

Table 5: The winning rates (%) of different methods on environment with changes in the underlying states. Zerg indicates that we trained on zerg_360 and executed on zerg_360_epo, and protoss indicates that we trained on protoss_360 and executed on protoss_360_epo.

### C.4.2 MORE HIGHLY DYNAMIC ENVIRONMENTS

To further explore RGP's performance in environments with higher randomness, we conducted experiments on the zerg_360_epo map. This map builds upon SMACv2 by introducing random masking of enemy observations for each agent. Specifically, in an episode, when an enemy is observed by the first agent, the first agent is guaranteed to observe it as normal. Other agents have a 50% chance of being unable to observe the enemy for the remainder of the episode. This significantly limits the agents' observation capabilities, further increasing environmental dynamics (for more details, refer to the SMACv2[2]). The experimental results are presented in Figure 9 (b). We observe that: (1) RGP outperforms other methods in environments with higher dynamics. This demonstrates RGP's scalability to highly dynamic environments. (2) PTDE performs poorly, likely because the inconsistency between states used during training and execution is further amplified in more complex settings. This magnification of inconsistency errors causes its performance to fall below RGP and QMIX.

### C.5 ROBUSTNESS

To explore the robustness of RGP under changes in the underlying states of the environment, we trained on zerg_360 and executed on zerg_360_epo. Similarly, we trained on protoss_360 and executed on protoss_360_epo. The results are shown in Table 5. We can observe that under such strict environmental constraints, QMIX is almost entirely unable to achieve any victories. In contrast, RGP retain a certain level of decision-making capability. This demonstrates that our approach main-

---

[2]https://arxiv.org/pdf/2212.07489

tains robustness in environments with latent state changes. Even when unexpected variations in state dynamics occur, our method continues to maintains a clear advantage over the baseline.

