# OpenReview forum: "Reconstruction-Guided Policy: Enhancing Decision-Making through Agent-Wise State Consistency"
_ICLR.cc/2025/Conference — ICLR 2025 Poster_

### Official Review · Reviewer_hsoz · 2024-11-01

**Soundness:** 3
**Presentation:** 1
**Contribution:** 2
**Rating:** 6
**Confidence:** 3

**Summary:**

The authors introduce a method, Reconstruction-Guided Policy (RGP), which consists of a decision module and a guidance module. The decision module uses a diffusion model to reconstruct a per-agent agent-wise state from an agent’s partial observation of the full environment state which captures the agent’s relationship to other agents. This reconstructed agent-wise state is then used to condition a policy for agent action selection. The guidance module uses an attention-based model to guide the reconstructed agent-wise states. This ensures that the information agents condition on during acting and training remain aligned which reduces error accumulation and improves learning.

**Strengths:**

* The approach to guide the reconstruction of the individual agent-wise states is interesting and using a diffusion model in reinforcement learning is also interesting.
* The authors conduct detailed ablation studies to disentangle which aspects of their method have the greatest impact on algorithm performance.
* The authors test their method in continuous and discrete action space environments.

**Weaknesses:**

* The paper has multiple grammatical errors and spelling mistakes.
* The illustrative example in Lines 71 - 80 is difficult to follow. I think the authors are trying to say that agent $x_1$ incorrectly assumes that agent $x_0$’s coverage is represented by the 2nd and 3rd elements in the state vector instead of the correct 3rd and 4th elements. Is this what the authors mean here? Since this is an example used to illustrate the necessity of the method, I think it can be explained more clearly.
* While it is beneficial that the authors benchmark on SMACv2 instead of only on SMAC it is still quite a limitation in the work that most of the conclusions regarding the method are drawn from results on SMAC. This together with only 3 continuous action space benchmarking tasks is a rather limited evaluation set and not very extensive as the authors mention in the paper.
* When there is improvement over a baseline it is slight and oftentimes within one measure of error of other methods. For example, in Table 1 RGP+HPN overlaps with the results of HPN-QMIX on all scenarios and RGP overlaps with QMIX, VDN or QPLEX on all scenarios.
* In Table 3, the error is not taken into account when computing the Performance Improving Ratio of one method over another. This sketches an incomplete picture since it only uses the mean. For a more statistically sound comparison, error propagation should be incorporated into these calculations using standard error propagation techniques. Similarly, for the Performance Retention Ratio of one method over another in Table 2.
* In Appendix C, the authors show that using an MLP to generate the per-agent state leads to similar performance as using a diffusion model. The MLP is less computationally expensive, why do the authors not use this?

**Questions:**

* There is no code available, the link the authors have provided seems to only have a readme containing no text except the work readme. Could the authors please fix this?
* Could the authors elaborate on how the guidance module shifts focus to agents instead of to observation features?
* Could the authors please elaborate on how the decentralised execution differs from centralised training with decentralised execution in Section 5.3?
* The term dimensional-wise state is used but not clearly defined. What do the authors mean by this and can the authors clearly contrast this with the agent-wise state which is a key concept in the work?

---

> ### Author Response · Authors · 2024-11-24
>
> ###### **Q1:** The paper has multiple grammatical errors and spelling mistakes.
> **R1:** Thank you for your suggestion. We have made corrections to address this issue.
>
> ###### **Q2:** The illustrative example in Lines 71 - 80 is difficult to follow. I think the authors are trying to say that agent $x_1$ incorrectly assumes that agent $x_0$’s coverage is represented by the 2nd and 3rd elements in the state vector instead of the correct 3rd and 4th elements. Is this what the authors mean here? Since this is an example used to illustrate the necessity of the method, I think it can be explained more clearly.
> **R2:** Thank you for your suggestion. This is not what we mean here. We have made adjustments in the introduction to make it more clearly. For more details, please refer to the Introduction of the manuscript.
>
>
> ###### **Q3:** While it is beneficial that the authors benchmark on SMACv2 instead of only on SMAC, it is still quite a limitation in the work that most of the conclusions regarding the method are drawn from results on SMAC. This together with only 3 continuous action space benchmarking tasks is a rather limited evaluation set and not very extensive as the authors mention in the paper.
> **R3:** Thank you for your suggestion and positive feedback.
> First of all, We would like to clarify that
>
> - **(a):** our conclusions are not solely drawn from results on SMAC, but rather reached it by comprehensively considering the results from both SMAC and SMACv2.
> For example, in Section 5.3, our original text is: "we compare the retention ratio of RGP with the PTDE and SIDiff, which also reconstruct the state for decision making, on the environment of SMAC and SMACv2, and illustrate the results in Table 2." And we observed that regardless of whether it was on SMAC or SMACv2 maps, our method consistently maintained higher performance retention ratios compared to PTDE and SIDiff. Therefore, we concluded that our method effectively improves performance retention ratios.
>
> - **(b):**  For the evaluation in continuous action space, existing works, such as [1,2,3], conducted experiments only in discrete spaces. In contrast, we performed experiments in both continuous and discrete spaces.
> To further address your concern, we conducted additional experiments on the zerg_90_20_vs_20 and zerg_360_epo maps of SMACv2.  The zerg_90_20_vs_20 involves 20 agents battling against 20 enemies under the zerg_90 configuration, where agents have a 90-degree field of view. The zerg_360_epo introduces random masking of observations for each agent, under the zerg_360 configuration, where agents have a 360-degree field of view. The experimental results are presented in the table below. The results show that our method (RGP) outperforms the baselines in both environments. This demonstrates the effectiveness of RGP in a broader range of environment settings. The corresponding content has been updated in Appendix C.4 of the manuscript.
>
> |         | RGP   |  PTDE  |  QMIX  |
> | --------   | :-----:  | :----:  | :----:  |
> | zerg_90_20_vs_20        |   12.8±3.3   |   1.0±1.0   |   6.5±5.7   |
> | zerg_360_epo        |    21.4±2.7    |  3.7±1.9  |    18.2±2.7   |
>
> [1] Chen, Yiqun, et al. "Ptde: Personalized training with distillated execution for multi-agent reinforcement learning." the Thirty-Third International Joint Conference on Artificial Intelligence.
>
> [2] Xu, Zhiwei, et al. "SIDE: State Inference for Partially Observable Cooperative Multi-Agent Reinforcement Learning." Proceedings of the 21st International Conference on Autonomous Agents and Multiagent Systems. 2022.
>
> [3] Zhou, Yihe, et al. "Is centralized training with decentralized execution framework centralized enough for MARL?." arXiv preprint arXiv:2305.17352 (2023).
>
>
> ###### **Q4:** When there is improvement over a baseline it is slight and oftentimes within one measure of error of other methods. For example, in Table 1 RGP+HPN overlaps with the results of HPN-QMIX on all scenarios and RGP overlaps with QMIX, VDN or QPLEX on all scenarios.
> **R4:** Thank you for your feedback. Our method achieved relatively good performance, for example, in Table 1, the win rate of RGP is 73.1±4.6, while QMIX achieves only 38.1±28.3 in 6h_vs_8z. Besides, as shown in Table 3 , under the Protoss_90 setting, the results show that Qmix achieved 27.6±8.2, while RGP achieved 37.2±6.5, representing a performance improvement of 9.6.
>
> ###### **Q5:** In Table 3, the error is not taken into account when computing the Performance Improving Ratio of one method over another. This sketches an incomplete picture since it only uses the mean. For a more statistically sound comparison, error propagation should be incorporated into these calculations using standard deviation propagation techniques. Similarly, for the Performance Retention Ratio of one method over another in Table 2.
> **R5:** Thank you for your valuable suggestion. We have updated the corresponding data in the paper.

---

> > ### Author Response · Authors · 2024-11-24
> >
> > #### **Q6:** In Appendix C, the authors show that using an MLP to generate the per-agent state leads to similar performance as using a diffusion model. The MLP is less computationally expensive, why do the authors not use this?
> > **R6:** Thank you for your feedback. We choose diffusion here because it is particularly well-suited for modeling complex data distributions and has demonstrated superior performance in tasks across task of CV, NLP, and decision-making. We assume the underlying reason for MLP achieving comparable performance with diffusion may stem from our replacement of the U-Net architecture in the original DDPM paper [4] with an MLP, due to computational resource limitations. Nonetheless, as shown in Figure 7, diffusion still outperforms VAE, MLP, and GAN in most cases.
> >
> > [4] Ho, Jonathan, Ajay Jain, and Pieter Abbeel. "Denoising diffusion probabilistic models." Advances in neural information processing systems 33 (2020): 6840-6851.
> >
> >
> > #### **Q7:** There is no code available, the link the authors have provided seems to only have a readme containing no text except the work readme. Could the authors please fix this?
> > **R7:** Thank you for your feedback. We have checked that and uploaded to the anonymous repository.
> >
> > #### **Q8:** Could the authors elaborate on how the guidance module shifts focus to agents instead of to observation features?
> > **R8:** Thank you for your feedback. First, we split the global information by agent, breaking it down into information for each individual agent. Subsequently, in the attention mechanism, we treat each agent as a whole and perform computations at the agent level. This allows us to shift attention to the individual agents, rather than specific dimensions of the global state.
> >
> > #### **Q9:** Could the authors please elaborate on how decentralized execution differs from centralized training with decentralized execution in Section 5.3?
> > **R9:** Thank you for your feedback. Decentralized execution and centralized training with distributed execution(CTDE) are essentially the same.  We assume that you may have questions about the calculation of the performance retention rate in Table 2 of Section 5.3.  The performance retention rate(PRR) was introduced by PTDE [5], which could be calculated by: dividing the win rate of CTDE by the win rate of CTCE(centralized training with centralized execution). The difference between CTCE and CTDE is that CTCE has access to the global information used during the training process. To clarify the related explanation, we have updated the table 2 of manuscript.
> >
> > [5] Chen, Yiqun, et al. "Ptde: Personalized training with distillated execution for multi-agent reinforcement learning." the Thirty-Third International Joint Conference on Artificial Intelligence.
> >
> > #### **Q10:** The term dimension-wise state is used but not clearly defined. What do the authors mean by this and can the authors clearly contrast this with the agent-wise state which is a key concept in the work?
> > **R10:** Thank you for your feedback. Dimension-wise state refers to the fact that other methods focus on global representations while overlooking the inherent relationships between agents. They simply map the dimensionality of global state information without considering these agent-specific connections. Agent-wise state refers to the process where we decompose the global information into individual agent information. Correspondingly, we have updated the introduction.

---

> > > ### Comment · Reviewer_hsoz · 2024-11-25
> > >
> > > Thank you for taking the time to reply and for the updates to the manuscript. Please see my replies below.
> > >
> > > **A1 and A2**: Thank you for updating this.
> > >
> > > **A3 and A4**:  Thank you for raising this point. But again, if we compare RGP to all other baselines it has overlap with nearly all of them. 6h_vs_8z is the only map where RGP has statistically significant improvement over the baselines in Table 1. This illustrates my point that SMAC and SMACv2 are a limited environment set to use and claim that extensive benchmarks were done. In the table given in **R3** there is overlap between QMIX and RGP on both maps.
> > >
> > > **A5**: Thank you for updating this.
> > >
> > > **A6**: It seems that diffusion has statistical overlap with an MLP in all cases in Figure 7? Is it possible to elaborate on the computational resource constraint leading to replacing the U-Net with an MLP?
> > >
> > > **A7 - A10**: Thank you for updating and clarifying these points.

---

> > > > ### Author Response · Authors · 2024-11-26
> > > >
> > > > Thank you for your timely response! We are also glad that our responses have addressed some of your concerns. We greatly appreciate you letting us know that some of your concerns have not been fully addressed. Here are our responses:
> > > >
> > > >
> > > > ###### **Q11:**  "If we compare RGP to all other baselines it has overlap with nearly all of them. 6h_vs_8z is the only map where RGP has statistically significant improvement over the baselines in Table 1. This illustrates my point that SMAC and SMACv2 are a limited environment set to use and claim that extensive benchmarks were done. In the table given in R3 there is overlap between QMIX and RGP on both maps."
> > > >
> > > > **R11:** It is true that RGP shows some overlap with the baselines in certain cases. We assume there are two reasons:
> > > >
> > > > (1) This could be due to the differences in random seed initialization, which causes the final results to fluctuate around the mean. This is also a common issue in most existing methods [5, 6, 7, 8].
> > > >
> > > > (2) This is because the results in Table 1 were obtained from SMAC and SMACv2 with 360-degree observation.  As you kindly pointed out, compared with the environment which is dynamically changing, the environments in Table 1 are less challenge (even though prior works mostly conducted experiments in the majority of SMAC and a few SMACv2 environments).
> > > >
> > > > To address this, we have conducted further experiments in Sections 5.6, and Appendix C.5.
> > > >
> > > > - **(a):** In Section 5.6, we tested the performance of RGP in environments with continuous action spaces, where it outperforms the baselines across all cases. **Importantly**,  As shown in Figure 5(b), RGP outperforms the baselines significantly, with the average reward for RGP reaching 1600, while the best baseline performance is around 1200.
> > > >
> > > > - **(b):** In Appendix C.5, we further validate the effectiveness of our method when latent environmental dynamics change. Specifically, we trained on zerg_360 and executed on zerg_360_epo. Similarly, we trained on protoss_360 and executed on protoss_360_epo. The results are shown in the table below, where "Zerg" indicates that we trained on zerg_360 and executed on zerg_360_epo, and "Protoss" indicates that we trained on protoss_360 and executed on protoss_360_epo. We observe that under such strict environmental constraints, QMIX is almost entirely unable to achieve any victories. In contrast, RGP retains a certain level of decision-making capability. This demonstrates that our approach remains effective in environments with latent state changes.
> > > >
> > > > |         | RGP   |  QMIX  |
> > > > | --------   | :-----:  | :----:  |
> > > > | zerg        |   7.4±4.5   |   2.7±1.9   |
> > > > | protoss        |   14.2±4.2     |  1.6±0.1  |
> > > >
> > > > - **(c):** To further validate the effectiveness of the RGP method over baseline methods, we performed a paired t-test comparing the results of RGP and QMIX across all environments in Table 3, and another t-test comparing the results of RGP+HPN and HPN-QMIX across the same environments. The results are presented in the table below.
> > > > From the table, we can observe that the p-values for both comparisons are below 0.05, which indicates that our method outperforms the baseline methods overall, further demonstrating the effectiveness of our approach.
> > > >
> > > > |      | RGP with QMIX| RGP+HPN with HPN-QMIX|
> > > > | --------   | :-----:  | :----:  |
> > > > | p |   0.0003  |   0.0010 |
> > > >
> > > > In summary, while RGP's advantages may not have been fully evident in Table 1, the subsequent experiments confirm that RGP is indeed an effective method.
> > > >
> > > > [5] Chen, Yiqun, et al. "Ptde: Personalized training with distillated execution for multi-agent reinforcement learning." the Thirty-Third International Joint Conference on Artificial Intelligence.
> > > > [6] Papoudakis, Georgios, Filippos Christianos, and Stefano Albrecht. "Agent modelling under partial observability for deep reinforcement learning." Advances in Neural Information Processing Systems 34 (2021): 19210-19222.
> > > > [7] Yang, Yaodong, et al. "Transformer-based working memory for multiagent reinforcement learning with action parsing." Advances in Neural Information Processing Systems 35 (2022): 34874-34886.
> > > > [8] Jing Sun, Shuo Chen, Cong Zhang, Yining Ma, and Jie Zhang. Decision-making with speculative opponent models. IEEE Transactions on Neural Networks and Learning Systems, 2024.

---

> > > > > ### Author Response · Authors · 2024-11-26
> > > > >
> > > > > ###### **Q12:** " It seems that diffusion has statistical overlap with an MLP in all cases in Figure 7? "
> > > > >
> > > > > **R12:** Thank you for your comment.
> > > > > we would like to clarify that the **diffusion model** is not an indispensable component in our framework. It is simply one of the tools used for state reconstruction, and it can be replaced with other generative models. We chose the diffusion model primarily because of its strong distribution modeling capabilities, which have been demonstrated in fields like computer vision and decision-making. Therefore, it holds particular promise when handling more complex environments with dynamic changes, where its ability to capture such distributions is especially valuable.
> > > > >
> > > > > As you kindly mentioned, the **MLP setting** demonstrates outstanding performance, and these results also highlight the effectiveness of our reconstruction-guided policy—even with the simplest state reconstruction network, it still delivers strong performance.
> > > > >
> > > > > ###### **Q13:** " Is it possible to elaborate on the computational resource constraint leading to replacing the U-Net with an MLP?"
> > > > >
> > > > > **R13:** We conducted a resource consumption analysis for training on the **zerg_360 map** under different settings. The settings we evaluated are as follows：
> > > > >
> > > > > **MLP:** We use a simple MLP network as the generative model in the reconstruction module to generate the reconstructed agent-wise states.
> > > > >
> > > > > **MLP-based diffusion:** we use a diffusion with MLP architecture as the generative model in the reconstruction module.
> > > > >
> > > > > **U-Net-based diffusion:** we use a diffusion  with U-Net architecture as the generative model in the reconstruction module.
> > > > > The corresponding results are:
> > > > > |         | MLP |  MLP-based diffusion |   U-Net-based diffusion |
> > > > > | --------   | :-----:  | :----:  | :----:  |
> > > > > | time consumption |   20 hours  |   2 days   |  5 days   |
> > > > > | GPU memory usage  |   5,536 MB     |  22,602 MB  | 28,293 MB  |
> > > > > |CPU memory usage   |    600% |   800% |   1200% |
> > > > >
> > > > > From this comparison, we observed that the **U-Net-based diffusion setting** required the most time and consumed the most resources. Since our GPU and CPU are shared within our research team, in order to minimize the impact on other experiments and reduce training time, we replaced the U-Net in diffusion with **MLP**. This change allowed us to complete the experiment more efficiently while still maintaining reasonable performance.
> > > > >
> > > > > [4] Ho, Jonathan, Ajay Jain, and Pieter Abbeel. "Denoising diffusion probabilistic models." Advances in neural information processing systems 33 (2020): 6840-6851.
> > > > >
> > > > > **Thanks again for your time and effort in reviewing our paper!**

---

> > > > > > ### Author Response · Authors · 2024-11-28
> > > > > >
> > > > > > Dear Reviewer hsoz,
> > > > > >
> > > > > > We have provided detailed responses to your concerns two days ago, we hope these responses have adequately addressed your concerns. As time goes by, we sincerely request your further responses.
> > > > > >
> > > > > > If we have resolved your issues, please consider raising your score to the positive side. If you have any further questions, please feel free to share them with us! Any valuable feedback is crucial for improving our work.
> > > > > >
> > > > > > Best regards,
> > > > > >
> > > > > > The Authors

---

> > > > > > > ### Comment · Reviewer_hsoz · 2024-12-02
> > > > > > >
> > > > > > > Thank you for your replies, the extra statistical analysis and for addressing my concerns.
> > > > > > >
> > > > > > > I will increase my score to a 6.

---

> ### Author Response · Authors · 2024-12-03
>
> **Thank you for your thoughtful consideration and for raising your score. We're glad that our revisions addressed your concerns. Your support means a lot to us!**

---

### Official Review · Reviewer_qsBB · 2024-11-01

**Soundness:** 3
**Presentation:** 2
**Contribution:** 2
**Rating:** 5
**Confidence:** 4

**Summary:**

The paper introduces Reconstruction-Guided Policy (RGP), a method in cooperative multi-agent reinforcement learning (MARL) to address partial observability, where agents rely on limited, localized information instead of global state access. Traditional MARL approaches train with global state access but deploy with reconstructed state representations, often reducing robustness and performance. RGP is evaluated on both discrete (SMAC, SMACv2) and continuous (predator-prey, cooperative navigation) environments, demonstrating superior performance and higher retention rates compared to methods like PTDE and SIDiff during centralized training and decentralized execution. Ablation studies and retention analysis further validate RGP’s effectiveness in preserving agent dependencies and decision consistency, showcasing its versatility in various MARL scenarios with observability challenges.

**Strengths:**

1. Proposes an innovative approach for state consistency in MARL, which mitigates errors from training-execution state inconsistencies.

2. Effectively captures inter-agent relationships, improving collaborative performance.

3. Demonstrates versatility and strong performance in both discrete and continuous action environments through extensive experiments.

**Weaknesses:**

1. Limited exploration of potential biases introduced by the generative model in state reconstruction, particularly in complex environments.
2. The approach’s scalability to more agents and highly dynamic environments is not thoroughly evaluated.
3. The theoretical foundation for using diffusion models over other generative models could be further clarified, as comparable methods like VAE and MLP produced similar results.
4. Misleading use of letters/symbols, unclear annotations on some images, poor writing and reading experience

**Questions:**

1. The paper uses $\epsilon$ repeatedly across equations, specifically in Eq (3) and Eq (6). Could you clarify why the same $\epsilon$ notation is applied in both contexts? Given that Eq (3) refers to a noise variable for the diffusion process while Eq (6) applies to the exploration parameter, distinguishing between these variables would improve clarity and reduce confusion.
2. Figure 2 is challenging to interpret with respect to the network design for RGP. Could you provide a more detailed explanation of how each component in Figure 2 maps to the decision and guidance modules described in Sections 4.1 and 4.2? Specifically, illustrating the data flow within and between these modules and explaining how key operations like the generative model, RNN, and attention mechanisms are integrated would be very helpful.
3. Could you elaborate on the robustness of the learned policies when deployed in environments with potential state changes? How well does RGP handle discrepancies between the training and execution environments, especially if the state dynamics change unexpectedly? Additionally, have you considered incorporating techniques like domain randomization or online adaptation to improve robustness in dynamic deployment settings?
4. The large standard deviations observed in baseline methods like SIDiff (Table 2) and QPLEX (Table 1, 3s5z vs 3s6z) suggest significant variability in these baselines’ performance. Could you clarify the potential sources of this variability? Additionally, could you explain the rationale behind selecting baselines with such high variance, as they may impact the reliability of comparative assessments?
5. While the paper uses a diffusion model for reconstructing agent-wise states, the ablation study in Figure 6 suggests that alternative models (e.g., VAE, MLP) yield similar performance. Could you provide further justification for the choice of the diffusion model over simpler architectures, especially given the increased computational complexity?
6. The paper evaluates RGP in multi-agent environments with relatively small agent counts. How scalable is the proposed approach in scenarios with a significantly larger number of agents, where maintaining agent-wise state consistency could become computationally intensive?

---

> ### Author Response · Authors · 2024-11-24
>
> #### **Q1:** Limited exploration of potential biases introduced by the generative model in state reconstruction, particularly in complex environments.
> **R1:** Thank you for your suggestion. In response, we analyzed the bias between the reconstructed states and the ground truth of the state (hereafter referred to as the true state) on the zerg_90 map of SMACv2, where 90 indicates that the agents have a 90-degree field of view. We calculated the cosine similarity between the reconstructed state and the true state, with the results illustrated in Figure 8 of Appendix C.3. For more details, please refer to the manuscript. From the figure, it can be observed that the state reconstructed by our method is significantly closer to the true state compared to the baseline. Furthermore, the similarity between the state reconstructed by our method and the true state consistently exceeds 0.9, demonstrating that our method introduces minimal bias during reconstruction.
>
> #### **Q2:** The approach's scalability to more agents and highly dynamic environments is not thoroughly evaluated.
> **R2:** Thank you for your feedback. Compared to most previous works [1,2,3,4,5,6,7], our work was evaluated in environments with more agents and highly dynamic environments. Most existing studies[1,2,3,4,5,6,7] use the SMAC environment to conduct experiments, involving less than 20 agents. In contrast, as Table 1 illustrates, we evaluated our method on the SMAC map 27m_vs_30m, which features the largest number of agents (involving 27 agents) in SMAC. Moreover, we also conducted experiments in SMACv2, which introduces additional high dynamics compared to SMAC.
>
> To further address your concern, we conducted additional experiments on the zerg_90_20_vs_20 and zerg_360_epo maps of SMACv2. The zerg_90_20_vs_20 involves 20 agents battling against 20 enemies under the zerg_90 configuration, where agents have a 90-degree field of view. The zerg_360_epo introduces random masking of observations for each agent, under the zerg_360 configuration, where agents have a 360-degree field of view. The experimental results are presented in the table below. The results show that our method (RGP) outperforms the baselines in both environments. This demonstrates RGP's scalability to environments with more agents and high dynamics. The corresponding content has been updated in Appendix C.4 of the manuscript.
> And please refer to SMACv2[8] for more details about zerg_360_epo.
>
> |         | RGP   |  PTDE  |  QMIX  |
> | --------   | :-----:  | :----:  | :----:  |
> | zerg_90_20_vs_20        |   12.8±3.3   |   1.0±1.0   |   6.5±5.7   |
> | zerg_360_epo        |    21.4±2.7    |  3.7±1.9  |    18.2±2.7   |
>
> [1] Chen, Yiqun, et al. "Ptde: Personalized training with distillated execution for multi-agent reinforcement learning." the Thirty-Third International Joint Conference on Artificial Intelligence.
>
> [2] Xu, Zhiwei, et al. "SIDE: State Inference for Partially Observable Cooperative Multi-Agent Reinforcement Learning." Proceedings of the 21st International Conference on Autonomous Agents and Multiagent Systems. 2022.
>
> [3] Yang, Yaodong, et al. "Transformer-based working memory for multiagent reinforcement learning with action parsing." Advances in Neural Information Processing Systems 35 (2022): 34874-34886.
>
> [4] Wang, Jiangxing, Deheng Ye, and Zongqing Lu. "More Centralized Training, Still Decentralized Execution: Multi-Agent Conditional Policy Factorization." The Eleventh International Conference on Learning Representations.
>
> [5] Zhou, Yihe, et al. "Is centralized training with decentralized execution framework centralized enough for MARL?." arXiv preprint arXiv:2305.17352 (2023).
>
> [6] Qihan Liu, Jianing Ye, Xiaoteng Ma, Jun Yang, Bin Liang, and Chongjie Zhang. Efficient multi-agent reinforcement learning by planning. arXiv preprint arXiv:2405.11778, 2024.
>
> [7] Jing Sun, Shuo Chen, Cong Zhang, Yining Ma, and Jie Zhang. Decision-making with speculative opponent models. IEEE Transactions on Neural Networks and Learning Systems, 2024.
>
> [8] Benjamin Ellis, Jonathan Cook, Skander Moalla, Mikayel Samvelyan, Mingfei Sun, Anuj Mahajan,Jakob Foerster, and Shimon Whiteson. Smacv2: An improved benchmark for cooperative multi-agent reinforcement learning. Advances in Neural Information Processing Systems, 36, 2024.

---

> ### Author Response · Authors · 2024-11-24
>
> #### **Q3:** The theoretical foundation for using diffusion models over other generative models could be further clarified, as comparable methods like VAE and MLP produced similar results.
> **R3:** Thank you for your feedback. Our original intent of applying diffusion is to generate the agent-wise state, rather than refine diffusion. Therefore,  diffusion just serves as one module in our method. Theoretically speaking, any generative model, including VAE, GAN, or even simple MLP, can replace the diffusion in our method. We choose diffusion here just because it is particularly well-suited for modeling complex data distributions and has demonstrated superior performance in tasks across CV, NLP, and decision-making.
> We assume the underlying reason for VAE and MLP achieving comparable performance with diffusion may stem from our replacement of the U-Net architecture in the original DDPM paper [9] with an MLP, due to computational resource limitations. Nonetheless, as shown in Figure 7, diffusion still outperforms VAE, MLP, and GAN in most cases.
>
> [9] Ho, Jonathan, Ajay Jain, and Pieter Abbeel. "Denoising diffusion probabilistic models." Advances in neural information processing systems 33 (2020): 6840-6851.
>
> #### **Q4:** Misleading use of letters/symbols, unclear annotations on some images, poor writing and reading experience.
> **R4:** Thank you for your comment. We have corrected the issues you pointed out in the manuscript, clarifying the image representations and standardizing the use of letters and symbols.
>
> #### **Q5:** The paper uses $\epsilon$ repeatedly across equations, specifically in Eq (3) and Eq (6). Could you clarify why the same $\epsilon$ notation is applied in both contexts? Given that Eq (3) refers to a noise variable for the diffusion process while Eq (6) applies to the exploration parameter, distinguishing between these variables would improve clarity and reduce confusion.
> **R5:** Thank you for your suggestion. We have revised the manuscript and represent the noise involved in the diffusion process as $\zeta$, while the exploration parameter for the greedy algorithm is denoted as $\epsilon$.
>
> #### **Q6:** Figure 2 is challenging to interpret with respect to the network design for RGP. Could you provide a more detailed explanation of how each component in Figure 2 maps to the decision and guidance modules described in Sections 4.1 and 4.2? Specifically, illustrating the data flow within and between these modules and explaining how key operations like the generative model, RNN, and attention mechanisms are integrated would be very helpful.
> **R6:** Thank you for your feedback. We have updated the legend to Figure 2 and provided a more detailed explanation. The integration of key operations is as follows.
>
> The decision module is active during both training and execution. At each timestep, the local observations $o_t^i$ and reconstructed trajectory states $h_{t-1}^i$ (computed from the previous time step t-1) are input into the diffusion model to generate the reconstructed agent-wise state $s_t^i$. Subsequently, the reconstructed agent-wise state $s_t^i$, local observations $o_t^i$, and reconstructed trajectory states $h_{t-1}^i$ are processed by RNN and MLP to estimate the reconstructed action value $Q_i$ and the reconstructed trajectory states $h_{t}^i$. The action value $Q_i$ is used to decide which action the agent takes. Besides, all reconstructed action values are optimized using a mixing network via TD loss.
>
> Considering that the diffusion-based generation process lacks ground truth for training, we design a guidance module, which is active only during training. Attention mechanisms in the guidance module take local observations $o_t^i$ and the decomposed global state $\bar{\boldsymbol{S}}_t$ as inputs, producing agent-wise states $\bar{s}_t^i$, serving as ground truth for generation. Afterward, the agent-wise state $\bar{s}_t^i$, local observations $o_t^i$, and trajectory states $\bar{h}_t^i$ are used by the RNN to estimate action values. Similarly, all action values $\bar{Q}_i$ are optimized using the same mixing network. This approach helps us obtain high-quality agent-wise states, which in turn guide the reconstruction process of diffusion.

---

> > ### Author Response · Authors · 2024-11-24
> >
> > #### **Q7:** Could you elaborate on the robustness of the learned policies when deployed in environments with potential state changes? How well does RGP handle discrepancies between the training and execution environments, especially if the state dynamics change unexpectedly?
> > **R7:**  Thank you for your comment. To investigate the robustness of the learned policy under unexpected state dynamics, we conducted additional experiments. Specifically, to simulate variations in the potential states of the environment, we trained RGP on protoss_360 and tested it on protoss_360_epo, as well as trained RGP on zerg_360 and tested it on zerg_360_epo. EPO represents a highly challenging setting that builds upon standard SMACv2 by introducing additional observation masking for agents. Specifically, during an episode, if an enemy is observed for the first time by one agent, that agent can continue observing the enemy as usual. However, other agents have a 50% probability of never being able to observe that enemy. This significantly limits the agents' observation capabilities, further increasing environmental dynamics. The results are shown in the table below, from which we can observe that baseline method have completely failed to achieve any victories, whereas our approach still maintains a certain win rate, demonstrating the robustness of our method. The relevant experiments have been added in Appendix C.5.
> >
> >   |           | zerg|  protoss|
> >   | :----:   | :-----:  | :----:  |
> >   | RGP  |    7.4±4.5    |   14.2±4.2  |
> >  | QMIX  |    2.7±1.9    |    1.6±0.1  |
> >
> >
> > #### **Q8:** Have you considered incorporating techniques like domain randomization or online adaptation to improve robustness in dynamic deployment settings?
> > **R8:** Thank you very much for your suggestion. Domain randomization and online adaptation are indeed intriguing ideas. However, domain randomization often requires complex parameter tuning, and online adaptation tends to face stability issues in complex and dynamic environments. Therefore, we are not currently considering incorporating domain randomization or online adaptation methods. We believe these could be promising directions for future research.
> >
> > #### **Q9:** The large standard deviations observed in baseline methods like SIDiff (Table 2) and QPLEX (Table 1, 3s5z vs 3s6z) suggest significant variability in these baselines' performance. Could you clarify the potential sources of this variability? Additionally, could you explain the rationale behind selecting baselines with such high variance, as they may impact the reliability of comparative assessments?
> > **R9:** Thank you for your feedback.
> >
> > For **the reason for the large standard deviation**, we believe that the high variance of SIDiff is caused by the inconsistency between the states used during training and execution. Specifically, it uses the true global state to assist the decision network in selecting actions during training but relies on the reconstructed global state during execution. The original reconstruction error might be further amplified by the decision network, especially in dynamically changing environments, leading to instability in the results. The high standard deviation of QPLEX is attributed to its dual dueling mixing network, which enhances representation capability and expands the exploration space. The increased exploration space makes it more difficult to converge in certain environments, such as corridor and zerg_360, leading to higher instability.
> >
> > As for **the reason of selecting them as baselines**, we chose SIDiff as a baseline as it represents a recent advancement in this field, published in September 2024. It uses diffusion to directly predict the global state, addressing partial observability, which is related to our work. We selected QPLEX as a baseline because it is widely adopted in existing multi-agent reinforcement learning methods, such as [10, 11, 12], and has demonstrated good performance in environments like protoss_360 and terran_360.
> >
> > [10] Jianye, H. A. O., et al. "Boosting multiagent reinforcement learning via permutation invariant and permutation equivariant networks." The Eleventh International Conference on Learning Representations. 2022.
> >
> > [11] Zang, Yifan, et al. "Automatic grouping for efficient cooperative multi-agent reinforcement learning." Advances in Neural Information Processing Systems 36 (2024).
> >
> > [13] Huang, Wenhan, et al. "Multiagent q-learning with sub-team coordination." Advances in Neural Information Processing Systems 35 (2022): 29427-29439.

---

> > > ### Author Response · Authors · 2024-11-24
> > >
> > > #### **Q10:** The paper evaluates RGP in multi-agent environments with relatively small agent counts. How scalable is the proposed approach in scenarios with a significantly larger number of agents, where maintaining agent-wise state consistency could become computationally intensive?
> > > **R10:** Thank you for your feedback.
> > > For the scalability of our RGP in scenarios with a significantly larger number of agents please refer Q2.
> > >
> > > For the computational burden in scenarios involving a larger number of agents, increasing the number of agents does not lead to an increase in the number of model parameters. This is because we employ a centralized training and distributed execution framework, where all agents share the same set of parameters. Centralized training allows for joint optimization of these shared parameters, while distributed execution ensures scalability during inference. Consequently, although the computational demands (e.g., inference and reconstruction overhead) may grow linearly with the number of agents, the resource requirements remain manageable due to parameter sharing.

---

> > > > ### Author Response · Authors · 2024-11-27
> > > > **Responses to Reviewer qsBB**
> > > >
> > > > Dear Reviewer qsBB,
> > > >
> > > > We have provided detailed responses to your concerns two days ago, we hope these responses have adequately addressed your concerns. As time goes by, we sincerely request your further responses.
> > > >
> > > > If we have resolved your issues, please consider raising your score to the positive side. If you have any further questions, please feel free to share them with us! Any valuable feedback is crucial for improving our work.
> > > >
> > > > Best regards,
> > > >
> > > > The Authors

---

> ### Comment · Reviewer_qsBB · 2024-11-30
> **Responses to Authors**
>
> In addressing **Q3**, the authors attempt to justify their choice of diffusion models by citing their theoretical advantages and superior performance in various tasks. However, this defense is severely undermined by a lack of clarity and consistency. The discrepancy between the reviewers' observations of similar performance across VAE, MLP, and diffusion models versus the authors' claim that diffusion outperforms the others in most cases remains inadequately addressed. The explanation regarding the replacement of the U-Net architecture with an MLP due to computational constraints is superficial and fails to convincingly demonstrate why diffusion models should still be preferred. Furthermore, the theoretical justification specific to generating agent-wise states is vague, leaving readers questioning the true merits of the chosen approach. The authors also neglect to provide a comprehensive comparison of computational efficiencies or scalability between diffusion models and the alternatives, which is crucial for validating their choice. This omission not only weakens their argument but also leaves a significant gap in understanding the practical implications of their methodology.
>
> Additionally, the authors neglect to address the broader concern about the overall poor writing and reading experience, focusing solely on symbols and annotations in their response to **Q4**. This oversight suggests that more extensive revisions are necessary to enhance the manuscript’s language and structure, which the authors have failed to adequately convey.
>
> The response to **Q7** is equally disappointing. Although the authors present additional experiments to demonstrate robustness, they fail to adequately contextualize the significance of the win rates reported. The improvement in win rates for RGP over QMIX is presented with standard deviations, yet there is no discussion on whether these improvements are statistically significant. The authors' response to **Q7** raises concerns regarding the robustness evaluation of their learned policies (RGP) in dynamic multi-agent environments. Specifically, the implementation of a 50% probability that other agents will never observe an enemy after one agent has observed it introduces substantial ambiguity and potential issues that undermine the validity of their robustness claims.
> In **Q9**, the authors attempt to explain the high variability observed in baseline methods like SIDiff and QPLEX, but their explanations remain speculative and lack supporting data or references. They attribute the high variance of SIDiff to inconsistencies between training and execution states and attribute QPLEX's high variance to its dual dueling mixing network expanding the exploration space. However, these explanations are not backed by empirical evidence or citations, making them appear unfounded. The selection of high-variance baselines is justified superficially, without adequately addressing how this impacts the reliability of their comparative assessments. This approach raises concerns about the validity of the comparative results presented, as the high variability could mask the true performance differences between methods.
>
> Upon examining the authors' response to **Q10**, it becomes clear that significant deficiencies remain unaddressed, further justifying the paper's marginal standing below the acceptance threshold. The response begins by deflecting the scalability inquiry back to **Q2**, which is not provided, leaving readers without a complete understanding of how scalability has been previously addressed. This omission demonstrates a lack of thoroughness and fails to engage directly with the reviewer’s concern about the proposed method's applicability to larger multi-agent environments. Furthermore, the authors assert that increasing the number of agents does not result in a rise in the number of model parameters due to their centralized training and distributed execution framework. While parameter sharing is a valid strategy for managing scalability, the response lacks empirical evidence or detailed analysis to substantiate the claim that computational demands remain manageable. The assertion that computational demands grow linearly with the number of agents is stated without supporting data, making it difficult to assess the practicality and efficiency of the proposed approach in real-world, large-scale scenarios.
>
> In summary, the paper suffers from inadequate presentation quality and numerous missed details that significantly detract from its overall impact and clarity. The authors' responses to the reviewers' comments are often vague, lacking the necessary specificity and depth to address the concerns raised convincingly. **I firmly think this paper is below the acceptance threshold and will keep the score.**

---

> ### Author Response · Authors · 2024-11-30
> **Response to Reviewer qsBB (part1)**
>
> Thank you for sharing your concerns with us. We also greatly appreciate the time and effort you have dedicated to reviewing our response. Regarding the points you raised, we believe there may have been a slight misunderstanding of our method, and we would like to clarify:
>
> **Q11:** "The discrepancy between the reviewers' observations of similar performance across VAE, MLP, and diffusion models versus the authors' claim that diffusion outperforms the others in most cases remains inadequately addressed. The explanation regarding the replacement of the U-Net architecture with an MLP due to computational constraints is superficial and fails to convincingly demonstrate why diffusion models should still be preferred."
>
> **R11:** Thank you for your comment. We would like to kindly clarify that diffusion is not the preferred, but rather optional. Diffusion is not an indispensable part of RGP; rather, it is the generative model that is truly essential. The core of RGP lies in solving the partial observability problem through agent-wise state reconstruction. Therefore, the key contribution of RGP is the framework for reconstructing agent-wise states, not the specific generative model used within the framework (since generative models are the work of other researchers, and are not proposed by RGP). As we mentioned in R3, any generative model, even an MLP, can be used for agent-wise state reconstruction in RGP. We simply selected a typical and representative generative model as the tool to implement state reconstruction, but this does not mean that diffusion is an indispensable component of RGP. Hence, which specific generative model performs best is not the primary focus of this paper, what is crucial is whether the RGP framework is effective. Therefore, we did not include the results of adapting different generative models to RGP in the main text of the manuscript, but rather placed them in the appendix Figure 7.
>
> The purpose of including Figure 7 in the appendix is to further explore which generative model adapts better to RGP, providing additional insights for readers when selecting a generative model for deploying RGP. The fact that the performance of MLP and VAE in Figure 7 is comparable to that of diffusion does not mean RGP is ineffective. On the contrary, the fact that it performs well even in the MLP setting highlights the effectiveness of the RGP framework, indicating that good performance can still be achieved even with the simplest generator configuration.
>
> **Q12:** "The implementation of a 50% probability that other agents will never observe an enemy after one agent has observed it introduces substantial ambiguity and potential issues that undermine the validity of their robustness claims."
>
> **R12:** Thank you for your feedback. The setting of a 50% probability that other agents will never observe an enemy after one agent has observed it is not our own implementation, but rather part of the official SMACv2 [1] setup.  Specifically, during an episode, if an enemy is observed for the first time by one agent, that agent can continue observing the enemy as usual. However, other agents have a 50% probability of never being able to observe that enemy during the rest of the episode. We consider this scenario a potential state change you mentioned, as the probability of agents observing the enemy introduces uncertainty and variability in their local states, depending on their prior observations. Due to this limited observation, agents may miss certain enemies and be unable to coordinate focused attacks with teammates, making them more vulnerable to enemy aggression. This significantly increases the challenge of the environment.
>
> The 50% probability can be replaced with any value, and the official documentation suggests that the lower the probability, the harder the task becomes. When the probability is set to 50%, models struggle to achieve modest performance according to the official document. Therefore, to show robustness and increase randomness, we chose to set the probability at 50%. We hope this addresses your concern.
>
> [1] Ellis, Benjamin, et al. "Smacv2: An improved benchmark for cooperative multi-agent reinforcement learning." Advances in Neural Information Processing Systems 36 (2024).

---

> ### Author Response · Authors · 2024-11-30
> **Response to Reviewer qsBB (Part2)**
>
> **Q13:** Although the authors present additional experiments to demonstrate robustness, they fail to adequately contextualize the significance of the win rates reported. The improvement in win rates for RGP over QMIX is presented with standard deviations, yet there is no discussion on whether these improvements are statistically significant.
>
> **R13:** To further demonstrate the effectiveness of our method from a statistical perspective, we performed a t-test on the results obtained by RGP and QMIX based on R7. The results are shown in the table below, where "Zerg" indicates that we trained on zerg_360 and executed on zerg_360_epo, and "Protoss" indicates that we trained on protoss_360 and executed on protoss_360_epo. We can observe that the p-values obtained for both settings are below 0.05, which statistically indicates that our method shows a significant improvement over the baseline methods. This demonstrates the robustness of our approach.
>
>   |           | zerg|  protoss|
>   | :----:   | -----:  | :----:  |
>   | RGP  |    7.4±4.5    |   14.2±4.2  |
>  | QMIX  |    2.7±1.9    |    1.6±0.1  |
> | p-value |   0.013  |   0.0006 |
>
> **Q14:**  In Q9, the authors attempt to explain the high variability observed in baseline methods like SIDiff and QPLEX, but their explanations remain speculative and lack supporting data or references.  This approach raises concerns about the validity of the comparative results presented, as the high variability could mask the true performance differences between methods.
>
> **R14:** Thank you for your feedback. We would like to further clarify：
>
> (1) The choice of baselines is reasonable. The widely accepted criteria for selecting baselines are popularity, effectiveness, and relevance. QPLEX is commonly used as a baseline in multi-agent reinforcement learning papers [2,3] (popularity) and performs well in some environments (effectiveness), such as protoss_360 and terran_360. On the other hand, SIDiff is highly relevant to our method (relevance) and was published in 2024. Therefore, we chose both of them as baselines.
>
> (2) The implementation of the experiments is correct. The code of QPLEX we used is the publicly available pymarl2 [4], with hyperparameters set according to those provided in the original paper to ensure optimal performance. SIDiff does not have public code, and we replicated it based on the details in the paper, using the hyperparameters specified in the original work. Moreover, QPLEX also exhibits high variance in experimental results in other papers (e.g., [2,3]). SIDiff, a more recent work, has not yet been widely cited, but the experimental results reported in the SIDiff paper [5] also show considerable variance in certain environments (e.g., Figures 3 and 4 in the SIDiff paper). Additionally, it should also be noted that while both methods show high variance in some maps, they do not exhibit high variance across all maps. Therefore, the high variance in both baselines likely arises from the limitations of the models in specific environments.
>
> (3) The analysis is of appropriate depth.  We have made some preliminary speculations on the reasons for the high variance of these two methods in certain environments (in R9). However, since our approach is not built upon these two models, nor is the focus of this paper on surveying the performance of various benchmarks, conducting further investigations, such as conducting experiments to explore the deep reasons behind the high variance of QPLEX and SIDiff, falls outside the scope of our work.
>
> (4) The high variance of QPLEX and SIDiff in certain environments does not affect the evaluation of RGP's performance. We can observe that in environments with high variance, neither QPLEX nor SIDiff serves as the best-performing baseline. Therefore, the high variance exhibited by these two models in their respective environments does not impact the evaluation of RGP's experimental performance.
>
>
> [2] Chen Y, Mao H, Zhang T, et al. Ptde: Personalized training with distillated execution for multi-agent reinforcement learning[J]. IJCAI2024
>
> [3] Huang, Wenhan, et al. "Multiagent q-learning with sub-team coordination." Advances in Neural Information Processing Systems 35 (2022): 29427-29439.
>
> [4] Hu, Jian, et al. "Rethinking the implementation tricks and monotonicity constraint in cooperative multi-agent reinforcement learning." arXiv preprint arXiv:2102.03479 (2021).
>
> [5] Xu, Zhiwei, et al. "Beyond local views: Global state inference with diffusion models for cooperative multi-agent reinforcement learning." arXiv preprint arXiv:2408.09501 (2024).

---

> ### Author Response · Authors · 2024-11-30
> **Response to Reviewer qsBB (Part3)**
>
> **Q15:** The response to Q10 begins by deflecting the scalability inquiry back to Q2, which is not provided, leaving readers without a complete understanding of how scalability has been previously addressed.
>
> **R15:** Thank you for your response. First, we would like to clarify that our RGP focuses on addressing the partial observability issue of agents, rather than the computational complexity faced by a large number of agents. Given the specific target of our method, we believe that not evaluating RGP in scenario involving a large number of agents does not significantly impact the validity of our approach.
>
> Next, for the scalability to a scenario with more agents, as we discussed in R2, most existing studies[2,6,7,8,9,10,11] use the SMAC environment to conduct experiments, involving less than 20 agents.  In contrast, as Table 1 illustrates, we evaluated our method on the SMAC map 27m_vs_30m, which features the largest number of agents (involving 27 agents) in SMAC. We also conducted additional experiments on the zerg_90_20_vs_20 maps of SMACv2. The zerg_90_20_vs_20 involves 20 agents battling against 20 enemies under the zerg_90 configuration, where agents have a 90-degree field of view. Compared to other methods, we have conducted experiments in environments with a larger number of agents.
>
> [2] Chen, Yiqun, et al. "Ptde: Personalized training with distillated execution for multi-agent reinforcement learning." the Thirty-Third International Joint Conference on Artificial Intelligence.
>
> [6] Xu, Zhiwei, et al. "SIDE: State Inference for Partially Observable Cooperative Multi-Agent Reinforcement Learning." Proceedings of the 21st International Conference on Autonomous Agents and Multiagent Systems. 2022.
>
> [7] Yang, Yaodong, et al. "Transformer-based working memory for multiagent reinforcement learning with action parsing." Advances in Neural Information Processing Systems 35 (2022): 34874-34886.
>
> [8] Wang, Jiangxing, Deheng Ye, and Zongqing Lu. "More Centralized Training, Still Decentralized Execution: Multi-Agent Conditional Policy Factorization." The Eleventh International Conference on Learning Representations.
>
> [9] Zhou, Yihe, et al. "Is centralized training with decentralized execution framework centralized enough for MARL?." arXiv preprint arXiv:2305.17352 (2023).
>
> [10] Qihan Liu, Jianing Ye, Xiaoteng Ma, Jun Yang, Bin Liang, and Chongjie Zhang. Efficient multi-agent reinforcement learning by planning. arXiv preprint arXiv:2405.11778, 2024.
>
> [11] Jing Sun, Shuo Chen, Cong Zhang, Yining Ma, and Jie Zhang. Decision-making with speculative opponent models. IEEE Transactions on Neural Networks and Learning Systems, 2024.
>
> **Q16:** While parameter sharing is a valid strategy for managing scalability, the response lacks empirical evidence or detailed analysis to substantiate the claim that computational demands remain manageable.  The assertion that computational demands grow linearly with the number of agents is stated without supporting data.
>
> **R16:** Thank you for your response.Since we use a homogeneous agent architecture where all agents share the same network, the number of agents can increase without increasing the network parameters. On the other hand, we employ a centralized training and distributed execution framework, where each agent's decision is made independently. Additionally, we only added modules in the agents to predict the agent-wise state without changing the parameters of the mixing network. As a result, the computational increase caused by diffusion is linear as the number of agents grows.
>
> We also conducted tests in the Predator-Prey environment, measuring the memory consumption when using 3, 6, and 9 agents. The results are shown in the table below. From the table, we can observe that as the number of agents increases, the GPU memory consumption grows linearly.
>
> |         | 3 agents|  6 agents |   9 agents |
> | --------   | :-----:  | :----:  | :----:  |
> | GPU memory usage  |   1576 MB     |  1750MB  | 1964 MB  |
>
>
> **Q17:**  "Additionally, the authors neglect to address the broader concern about the overall poor writing and reading experience, focusing solely on symbols and annotations in their response to Q4. This oversight suggests that more extensive revisions are necessary to enhance the manuscript's language and structure, which the authors have failed to adequately convey. "
>
> **R17:** We have conducted a thorough review and revision of the manuscript. In addition to the changes to the symbols and annotations you suggested, we have also provided further clarification in several descriptions.
> For example, to improve readability and understanding, we revised the legend in Figure 2 and adjusted its description to better reflect our main idea. We have also revised the description of the decision module in lines 234 to 240 to make it more fluent. We have made further revisions regarding tenses and spelling.

---

### Official Review · Reviewer_gRGP · 2024-11-02

**Soundness:** 3
**Presentation:** 2
**Contribution:** 2
**Rating:** 6
**Confidence:** 4

**Summary:**

Partial observability in multi-agent reinforcement learning (MARL) limits agents to local observations, often leading to suboptimal decisions and inconsistencies between training and execution. To solve this, the paper proposes the Reconstruction-Guided Policy (RGP), which reconstructs an agent-wise state capturing inter-agent relationships, ensuring consistent input for both training and execution. Experiments across discrete and continuous action environments show RGP’s good performance.

**Strengths:**

- The paper studies an important problem in cooperative MARL, that is dealing with partial observability and non-stationarity
- The paper proposes a novel method, which is based on state-of-the-art techniques such diffusion models and attention mechanism.
- The paper includes strong ablation study, which examines and justifies the selection of the method's components.
- The proposed method achieves good results across different benchmarks.

**Weaknesses:**

- The comparison with the related work can be improved: The paper ignores significant works based on agent (opponent) / state modelling, world models, transformer-based methods which also aim to deal with partial observability. For instance, the authors argue that one main challenge is state inconsistency. In particular, many works (e.g. [1]) leverage state embeddings (which can be much more compressed and informative) instead of using the reconstructed state (which can be less informative for each agent's training, also due to redundant information, see [2]). Therefore, although I believe that the proposed method is interesting, the state inconsistency challenge has partially been addressed by related work, and thus the authors should better highlight their contributions in the context of related work.
- One concern is that the baselines regarding partial observability are not very established in MARL community. Some of them have not even been published at a top-tier AI/ML venue. To further improve the paper, I would suggest that the authors also include well-known methods (as baselines) based on alternative ways to deal with partial observability (e.g. [3], [4], [5]).
- The attached repo does not contain any files.
- Minor weakness: The authors do not include information about hyperparameter details of the baselines.
- Minor weakness: There are many typos and grammatical errors (e.g. lines 240-244). I suggest that the authors proofread carefully the whole manuscript.

[1] Papoudakis, Georgios, Filippos Christianos, and Stefano Albrecht. "Agent modelling under partial observability for deep reinforcement learning." Advances in Neural Information Processing Systems 34 (2021): 19210-19222.

[2] Guan, Cong, et al. "Efficient multi-agent communication via self-supervised information aggregation." Advances in Neural Information Processing Systems 35 (2022): 1020-1033.

[3] Yang, Yaodong, et al. "Transformer-based working memory for multiagent reinforcement learning with action parsing." Advances in Neural Information Processing Systems 35 (2022): 34874-34886.

[4] Wen, Muning, et al. "Multi-agent reinforcement learning is a sequence modeling problem." Advances in Neural Information Processing Systems 35 (2022): 16509-16521.

[5] Sun, Jing, et al. "Decision-Making With Speculative Opponent Models." IEEE Transactions on Neural Networks and Learning Systems (2024).

**Questions:**

- How did the authors make the MPE' Navigation task (Spread) continuous ?
- What was the time horizon in SMAC tasks? Can the authors provide the plots from these experiments?

---

> ### Author Response · Authors · 2024-11-24
>
> #### **Q1:** The comparison with the related work can be improved: The paper ignores significant works based on agent (opponent) / state modelling, world models, transformer-based methods which also aim to deal with partial observability. For instance, the authors argue that one main challenge is state inconsistency. In particular, many works (e.g. [1]) leverage state embeddings (which can be much more compressed and informative) instead of using the reconstructed state (which can be less informative for each agent's training, also due to redundant information, see [2]). Therefore, although I believe that the proposed method is interesting, the state inconsistency challenge has partially been addressed by related work, and thus the authors should better highlight their contributions in the context of related work.
> **R1:** Thank you for your valuable suggestion.
> The other methods, when addressing state inconsistency issues, either overlook environmental information [1,3] or introduce compounded errors [4]. Our approach effectively avoids these problems.
>  We have revised our related work,  in which we highlighted the difference between our paper and the previous works [1,2]. Please refer to the manuscript for more details.
>
> [1] Papoudakis, Georgios, Filippos Christianos, and Stefano Albrecht. "Agent modelling under partial observability for deep reinforcement learning." Advances in Neural Information Processing Systems 34 (2021): 19210-19222.
>
> [2] Guan, Cong, et al. "Efficient multi-agent communication via self-supervised information aggregation." Advances in Neural Information Processing Systems 35 (2022): 1020-1033.
>
> [3] Jing Sun, Shuo Chen, Cong Zhang, Yining Ma, and Jie Zhang. Decision-making with speculative opponent models. IEEE Transactions on Neural Networks and Learning Systems, 2024.
>
> [4] Qihan Liu, Jianing Ye, Xiaoteng Ma, Jun Yang, Bin Liang, and Chongjie Zhang. Efficient multi-agent reinforcement learning by planning. arXiv preprint arXiv:2405.11778, 2024.
>
>
>
> #### **Q2:** One concern is that the baselines regarding partial observability are not very established in MARL community. Some of them have not even been published at a top-tier AI/ML venue. To further improve the paper, I would suggest that the authors also include well-known methods (as baselines) based on alternative ways to deal with partial observability.
> **R2:** Thank you for your suggestion. We conducted experiments in SMACv2 with three random seeds and compared our method with ATM[5]. The table below shows the mean and standard deviation. In the table, protoss_360 represents the setting where agents have a 360-degree field of view in the protoss scenario. We can observed that our results consistently outperform the baseline, demonstrating the effectiveness of our approach. The corresponding training curves have been incorporated into the manuscript.
> |         | Protoss_360   |  Terran_360  |  Zerg_360  |
> | --------   | :-----:  | :----:  | :----:  |
> | ATM        |   69.8±4.2   |   67.6±10.1   |    56.6±4.7   |
> | RGP        |    76.2±3.6    |  75.0±5.9  |   61.4±7.1   |
>
> [5] Yang, Yaodong, et al. "Transformer-based working memory for multiagent reinforcement learning with action parsing." Advances in Neural Information Processing Systems 35 (2022): 34874-34886.
>
> #### **Q3:** The attached repo does not contain any files.
> **R3:** Thank you for your feedback. We have checked it and updated the anonymous repository.
>
> #### **Q4:** Minor weakness: The authors do not include information about hyperparameter details of the baselines.
> **R4:** Thank you for your valuable suggestion. In Section 5.1, Implementation Details, we mentioned that all baseline methods were tuned using the hyperparameters from PyMARL2 [6]. Details about the hyperparameters of the baselines have been added to Appendix A2.
>
> [6] Hu, Jian, et al. "Rethinking the implementation tricks and monotonicity constraint in cooperative multi-agent reinforcement learning." arXiv preprint arXiv:2102.03479 (2021).

---

> > ### Author Response · Authors · 2024-11-24
> >
> > #### **Q5:** Minor weakness: There are many typos and grammatical errors (e.g. lines 240-244). I suggest that the authors proofread carefully the whole manuscript.
> > **R5:** Thank you for your suggestion. We have corrected this error and carefully proofread the manuscript to ensure our statements are accurate and precise.
> >
> > #### **Q6:** How did the authors make the MPE' Navigation task (Spread) continuous ?
> > **R6:** Thank you for your question. We assume you are asking how to make the action space continuous. We enable the continuous action space of PettingZoo's MPE by setting continuous_actions=True in the code . For more details, please refer to the official [MPE-PettingZoo](https://pettingzoo.farama.org/environments/mpe/) documentation. We have provided an additional explanation about this in Appendix B.2.
> >
> >
> > #### **Q7:** What was the time horizon in SMAC tasks? Can the authors provide the plots from these experiments?
> > **R7:** Thank you for your question. We assume by 'time horizon,' you are referring to the 'timesteps' in the training plots. We have provided the plots with the time horizon of the SMAC task in Appendix C.1. Refer to Appendix C.1 for more details.

---

> > > ### Author Response · Authors · 2024-11-27
> > >
> > > Dear Reviewer gRGP,
> > >
> > > We provided detailed responses to your concerns two days ago, and we sincerely hope that our responses have addressed them adequately. If you have any further questions or need additional clarification, please don’t hesitate to let us know. Your valuable feedback is essential to help us improve our work.
> > >
> > > Best regards,
> > >
> > > The Authors

---

> > > > ### Comment · Reviewer_gRGP · 2024-11-30
> > > >
> > > > I would like to thank the authors for their effort to address my concerns. However, I remain sceptical about the motivation of state-inconsistency. Why do [1] and [3] overlook environmental information? The augmented individual policies are conditioned on the latent agent-beliefs $z^i$ (similar to what the proposed method does) which have considered necessary environmental information.

---

> > > > > ### Author Response · Authors · 2024-12-01
> > > > >
> > > > > Thank you for sharing your concerns with us. We are also glad that our responses have addressed some of your concerns. We greatly appreciate you letting us know that some of your concerns have not been fully addressed. Here are our responses:
> > > > >
> > > > > **Q8:** I remain sceptical about the motivation of state-inconsistency.
> > > > >
> > > > > **R8:** State-inconsistency in our paper refers to the situation where the real state is used for decision-making during training, while the reconstructed state is used during execution, as overlooked in [1, 2]. Specifically, the policy is trained on the true state, but during execution, it operates on the reconstructed state, which may differ due to reconstruction inaccuracies. This inconsistency reduces the policy's effectiveness, as it was never trained to handle imperfect states.
> > > > >
> > > > > In contrast to [1, 2], our work makes decisions based on consistent  states (i.e., agent-wise state) during both training and execution. To be specific, no matter during training or execution, the agent makes decisions based on the reconstructed agent state, which allows the policy to have the consistent state as input. This approach helps prevent the issue of error amplification during the execution phase.
> > > > >
> > > > > [1] Chen, Yiqun, et al. "Ptde: Personalized training with distillated execution for multi-agent reinforcement learning." the Thirty-Third International Joint Conference on Artificial Intelligence.
> > > > >
> > > > > [2] Xu, Zhiwei, et al. "Beyond local views: Global state inference with diffusion models for cooperative multi-agent reinforcement learning." arXiv preprint arXiv:2408.09501 (2024).
> > > > >
> > > > > **Q9:** Why do [3] and [4] overlook environmental information? The augmented individual policies are conditioned on the latent agent-beliefs $z^i$ (similar to what the proposed method does) which have considered necessary environmental information.
> > > > >
> > > > > **R9:** Thank you for your feedback. The opponent modeling method used in [3, 4] primarily models other agents' actions, observations, and other agent-specific information, which are local to the agents themselves. However, the combination of local information from all agents is insufficient to fully represent the global state, even though it still influences decision-making. For instance, if there are obstacles in the environment that none of the agents have observed, these methods are unable to provide this obstacle information to the agents. Therefore, we believe that opponent modeling methods may overlook important environmental information.
> > > > >
> > > > > In contrast, our approach is based on the global state, rather than the local information of other agents. By starting from global information, it treats all potential environmental information as a single agent and assigns attention weights to each agent. This allows our method to consider all information that could potentially impact decision-making in a more comprehensive way.
> > > > >
> > > > > [3] Papoudakis, Georgios, Filippos Christianos, and Stefano Albrecht. "Agent modelling under partial observability for deep reinforcement learning." Advances in Neural Information Processing Systems 34 (2021): 19210-19222.
> > > > >
> > > > > [4] Jing Sun, Shuo Chen, Cong Zhang, Yining Ma, and Jie Zhang. Decision-making with speculative opponent models. IEEE Transactions on Neural Networks and Learning Systems, 2024.

---

> > > > > > ### Comment · Reviewer_gRGP · 2024-12-01
> > > > > >
> > > > > > I would like to thank the authors for their thoughtful feedback.
> > > > > >
> > > > > > I understand R8 and it is indeed helpful. **I believe that the proposed approach is interesting, and that is why I maintain my score for acceptance**.
> > > > > >
> > > > > > However, I still believe that [3] and [4] have already somewhat captured the state-inconsistency problem (motivation), because, in such methods, at time step $t$ the current agent's policy is explicitly conditioned on the history of observations, $h_t$, and the current agent's belief $z_t$, but since the policy network is recurrent, that implies that the policy is also conditioned on the previous agent beliefs $z_{\tau}$ (for  $\tau = 1, ..., t-1$) as well. Therefore, I suppose that in the example of the obstacle that the authors mentioned, each agent should have incorporated such information into their policy network, if some other agent had noticed the obstacle and also if that obstacle was important for the task.

---

> ### Author Response · Authors · 2024-12-02
>
> **Thank you for reviewing our paper and for sharing your concerns. Below are our responses to address them.**
>
> **Q10:** I still believe that [1] and [2] have already somewhat captured the state-inconsistency problem (motivation). I suppose that in the example of the obstacle that the authors mentioned, each agent should have incorporated such information into their policy network, if some other agent had noticed the obstacle and also if that obstacle was important for the task.
>
> **R10:** We agree with your opinion that [1] and [2] have captured the issue of state-inconsistency to some extent.
>
> However, compared to the state reconstruction-based method we proposed, they focus on the local observations and actions of each agent, without explicitly considering the importance of environmental information.
>
> As you mentioned, the information about the obstacle is recorded by the agent in the form of observation history $h_t$​. However, in [1], the recurrent network is not utilized, and therefore, there is no observation history. The method relies solely on the current observation $o_i^t$​ of agent $i$ to predict the actions of other agents, thereby overlooking potential environmental information.
> In [2], the gradients of the decision network are not backpropagated to the recurrent encoder. In other words, the observation history in the recurrent encoder is only used to reconstruct the current observations and actions of other agents. This limitation prevents the effective capture of comprehensive environmental information, potentially causing the agent to make suboptimal decisions.
>
> In contrast, our method explicitly incorporates the reconstruction of global information (agent-wise state) through the observation and observation history. This approach allows us to more effectively capture complete environmental information, thereby facilitating more efficient decision-making.
>
> Therefore, we believe that our improvements to the global state reconstruction method are both valuable and necessary.
>
> [1] Jing Sun, Shuo Chen, Cong Zhang, Yining Ma, and Jie Zhang. Decision-making with speculative opponent models. IEEE Transactions on Neural Networks and Learning Systems, 2024.
>
> [2] Papoudakis, Georgios, Filippos Christianos, and Stefano Albrecht. "Agent modelling under partial observability for deep reinforcement learning." Advances in Neural Information Processing Systems 34 (2021): 19210-19222.
>
>
> **Once again, thank you for your thoughtful review, as well as for taking the time to consider our additional results and visualizations. We appreciate your valuable feedback and the effort you have invested in evaluating our work. We sincerely hope that our responses have adequately addressed your concerns. Your support and insights are highly valuable and mean a great deal to us.**

---

### Official Review · Reviewer_aujm · 2024-11-04

**Soundness:** 4
**Presentation:** 4
**Contribution:** 3
**Rating:** 8
**Confidence:** 2

**Summary:**

This paper addresses two issues with multi-agent reinforcement learning: the errors from reconstructing the global state of the environment during execution, and respecting dependencies between agents which are necessary for optimal decision making. The paper shows how the model addresses these two issues and empirically dissect it to show its better performance than current MARL solutions, as well as reasons for what causes this.

**Strengths:**

* The writing flows well and is well-motivated. It was easy to understand the MARL background and the problems of state reconstruction error and inter-agent coordination that this paper tackles.
* Important assumptions and details (critical to the solution method and results) were made clear often to the reader: such as the guidance network only being used during the training, and the implementation details for the experiments.
* Enumerating the research questions tremendously helped with the paper's clarity and structure. This practice should be adopted into more papers in the field!

**Weaknesses:**

* Figures need a bit more clarity: What do the shaded regions in Fig 5 represent? How many runs were used for Figure 5? Depending on the number of seeds and the claim being made, this has to be specified as context to the reader. For example, tolerance intervals or bootstrapped confidence intervals are great for showing the variability in performance when there, even when there are a small number of runs. 95% confidence intervals would be appropriate if there are more runs and it's safe to assume the y-axis variable is normally distributed.
* The paper lacks a limitations section. Besides the one limitation regarding smaller fields of view, it  would be nice to see what other challenges to do with partial observability in MARL would need to be addressed next, and if there are any other shortcomings of RPG. Certain parts of the work speculated reasons about why results were seen (such as the impressive score of RGP+FACMAC in Predator-Prey). This could be summarized at the end of the conclusions section.
* [Very Minor] Line 483: typo “continuous”

**Questions:**

* What is the uncertainty measure in Table 1? Is it a standard error?

---

> ### Author Response · Authors · 2024-11-24
>
> #### **Q1:** Figures need a bit more clarity: What do the shaded regions in Fig 5 represent? How many runs were used for Figure 5? Depending on the number of seeds and the claim being made, this has to be specified as context to the reader.
> **R1:** Thank you for your comment. Following the settings of [1, 2, 3], we conducted the experiments shown in Figure 5 using three random seeds. Thick, dark-colored lines denote the mean, and the shaded areas represent standard deviations. We have clarified this explanation in the manuscript.
>
> [1] Liu, Qihan, et al. "Efficient Multi-agent Reinforcement Learning by Planning." The Twelfth International Conference on Learning Representations.
>
> [2] Ellis, Benjamin, et al. "Smacv2: An improved benchmark for cooperative multi-agent reinforcement learning." Advances in Neural Information Processing Systems 36 (2024).
>
> [3] Chen, Yiqun, et al. "Ptde: Personalized training with distillated execution for multi-agent reinforcement learning." the Thirty-Third International Joint Conference on Artificial Intelligence.
>
> #### **Q2:** The paper lacks a limitations section. Besides the one limitation regarding smaller fields of view, it would be nice to see what other challenges to do with partial observability in MARL would need to be addressed next, and if there are any other shortcomings of RPG. Certain parts of the work speculated reasons about why results were seen (such as the impressive score of RGP+FACMAC in Predator-Prey). This could be summarized at the end of the conclusions section.
> **R2:** Thank you for your valuable suggestions on our work. We have added the summary of the experiments and limitations in the section of conclusion.
>
>
> #### **Q3:** [Very Minor] Line 483: typo “continuous”.
> **R3:** Thanks for your comment! We have fixed this issue in our paper.
>
> #### **Q4:** What is the uncertainty measure in Table 1? Is it a standard error?
> **R4:** Thank you for your comment. Yes, you are absolutely right, the data in Table 1 represents the mean with standard deviation from three random seeds. We have added the description in Table 1 .

---

> > ### Author Response · Authors · 2024-11-27
> >
> > Dear Reviewer aujm,
> >
> > Thank you very much for your recognition of our work. We provided detailed responses to your concerns two days ago, and we sincerely hope that our responses have addressed them adequately. If you have any further questions or need additional clarification, please don’t hesitate to let us know. Your valuable feedback is essential to help us improve our work.
> >
> > Best regards,
> >
> > The Authors

---

> > > ### Comment · Reviewer_aujm · 2024-12-02
> > >
> > > Many thanks to the authors for swiftly addressing my concerns with the paper! After looking through the current draft, I am very happy with the state of the paper and the revisions made.

---

> > > > ### Author Response · Authors · 2024-12-02
> > > >
> > > > Thank you again for valuable suggestions and recognition of our contributions. We're glad that our revisions addressed your concerns. Your support means a lot to us!

---

### Author Response · Authors · 2024-12-04
**Global Response - Discussion Summary**

Dear PC, SAC, AC, and Reviewers,

We would like to express our sincere appreciation to the reviewers for their evaluations and constructive feedback throughout the discussion period. We appreciate the reviewers' recognition and thoughtful evaluation of our paper.

After extensive discussions, we have addressed the most of concerns raised by reviewers (aujm, gRGP, hsoz), and reviewers (aujm, gRGP, hsoz) have expressed a positive attitude toward our work, acknowledging the strengths of our work, including the **novelty** (gRGP, qsBB, hsoz), **strong performance** (gRGP, qsBB), and **extensive/detailed experiments and ablations** (acknowledged by all reviewers).

Possibly due to the busy schedule, reviewer qsBB did not provide feedback on our further responses, in which we have provided detailed experiments and responses. However, we believe they can resolve the issues raised by Reviewer qsBB.

Finally, we would like to express our sincere gratitude to the reviewers once again for their invaluable feedback, which has greatly contributed to improving the quality of our paper.

Best regards,

The Authors

---

### Meta-Review · Area_Chair_VTEu · 2024-12-20

**Metareview:**

This paper presents Reconstruction-Guided Policy (RGP), a technique to address the challenges of partial observability in the cooperative multi-agent reinforcement learning setting. RGP defines and reconstructs an agent-wise state, instead of the more common dimension-wise state in other techniques. This provides a mechanism to capture inter-agent relationships and helps maintain consistency between training and execution. The experiments provide evidence of RGP's performance on both discrete and continuous environments.

**Additional Comments On Reviewer Discussion:**

The authors provided detailed responses to the concerns raised by the reviewers. While most reviewers concur that the bar for acceptance has been reached, reviewer qsBB provided a dissenting opinion about some unresolved issues. The key disagreements seem to be about the empirical results and certain choices made during designing the experiments. Having carefully reviewed the discussion I feel that the authors have provided sufficient justification for their experimental choices and results (for example, the choice of 50% is an environment default, and a competitive level of scalability has been demonstrated by additional experiments). In particular, I do not think that the changes suggested by the reviewer would significantly impact the scientific contributions of this paper. As it stands, especially accounting for the changes already incorporated in the paper after the discussion period, the paper makes a significant enough contribution to warrant acceptance.

---

### Decision · Program_Chairs · 2025-01-22

Accept (Poster)